# Structural basis for two metal-ion catalysis of DNA cleavage by Cas12i2

Xue Huang[1,2,3], Wei Sun[2,3], Zhi Cheng[2,3,4], Minxuan Chen[2,3,4], Xueyan Li[2,3,4], Jiuyu Wang[2,3], Gang Sheng[3 ✉], Weimin Gong[1,5 ✉] & Yanli Wang [2,3,4 ✉]

To understand how the RuvC catalytic domain of Class 2 Cas proteins cleaves DNA, it will be necessary to elucidate the structures of RuvC-containing Cas complexes in their catalytically competent states. Cas12i2 is a Class 2 type V-I CRISPR-Cas endonuclease that cleaves target dsDNA by an unknown mechanism. Here, we report structures of Cas12i2–crRNA–DNA complexes and a Cas12i2–crRNA complex. We reveal the mechanism of DNA recognition and cleavage by Cas12i2, and activation of the RuvC catalytic pocket induced by a conformational change of the Helical-II domain. The seed region (nucleotides 1–8) is dispensable for RuvC activation, but the duplex of the central spacer (nucleotides 9–15) is required. We captured the catalytic state of Cas12i2, with both metal ions and the ssDNA substrate bound in the RuvC catalytic pocket. Together, our studies provide significant insights into the DNA cleavage mechanism by RuvC-containing Cas proteins.

[1] Hefei National Laboratory for Physical Sciences at the Microscales, University of Science and Technology of China, 230026 Hefei, Anhui, China. [2] National Laboratory of Biomacromolecules, CAS Center for Excellence in Biomacromolecules, Institute of Biophysics, Chinese Academy of Sciences, 100101 Beijing, China. [3] Key Laboratory of RNA Biology, CAS Center for Excellence in Biomacromolecules, Institute of Biophysics, Chinese Academy of Sciences, 100101 Beijing, China. [4] University of Chinese Academy of Sciences, 100049 Beijing, China. [5] School of Life Sciences, University of Science and Technology of China, 230026 Hefei, Anhui, China. ✉email: shenggang@ibp.ac.cn; wgong@ustc.edu.cn; ylwang@ibp.ac.cn

Clustered regularly interspaced short palindromic repeats (CRISPR)-Cas (CRISPR-associated proteins) is an RNA-guided adaptive immunity system present in bacteria and archaea, which defends its host against foreign nucleic acids that originate from phages or plasmids[1]. CRISPR-Cas systems are grouped into two classes, six types, and over 30 subtypes, based on the constitution difference and sequence divergence between the effector proteins[2,3]. Class 1 CRISPR-Cas systems comprise type I, III, and IV, while Class 2 CRISPR-Cas systems include type II, V, and VI. Class 2 CRISPR-Cas systems consist of a single large protein–crRNA complex for CRISPR interference, exhibiting programmable nuclease activities, which makes them suitable tools for genome-editing and nucleic acid detection[1,4,5]. Cas9, the signature protein of the type II system, cleaves the target and non-target strands by its HNH and RuvC catalytic domain. In contrast, Cas12a, the hallmark protein of the type V system, uses a single RuvC domain for cleavage of both strands of target DNA. Cas9 and Cas12a are the two Cas proteins that are used most often, and both encompass a RuvC catalytic domain[5–8]. However, structures of Cas9 or Cas12 in the active state that have both metal ions and DNA bound within the RuvC catalytic pocket have not yet been determined, although numerous structures of Cas9 and Cas12 have been determined[9–19]. To understand how the RuvC domain cleaves DNA, it is critical to elucidate the structures of RuvC-containing Cas complexes in their catalytically competent states, with both metal ions and ssDNA substrate bound in the RuvC catalytic pocket.

Type V is the most diversified type in class 2 CRISPR-Cas system and is further divided into subtypes A–K. Effectors Cas12a–Cas12k comprise a single RuvC catalytic domain but share little sequence conservation[20,21]. Cas12a (also termed Cpf1) is currently the best-studied Cas12 effector. Cas12a only requires a CRISPR RNA (crRNA) for its RNA-guided DNA cleavage activities via the RuvC domain. In contrast, Cas9 requires both a crRNA and a *trans*-activating CRISPR RNA (tracrRNA) for its activity[7,22]. In addition, Cas12a is also an RNase that is capable of processing the pre-crRNA to generate mature crRNA[23]. Unlike Cas12a, Cas12b (also termed C2c1) lacks pre-crRNA processing activity and requires both crRNA and tracrRNA for its DNA cleavage activity[24]. Cas12b contains a single RuvC domain, which cleaves the two strands of the target dsDNA sequentially[9].

Few studies are available for other subtypes of the type V CRISPR-Cas system. Recently, type V-C, -G, -H, and -I systems have been identified[21]. Two Cas12i (Cas12i1 and Cas12i2) proteins have crRNA-guided dsDNA cleavage activity. Cas12i proteins are distantly related to Cas12b based on phylogeny but resemble Cas12a in the fact that it also relies on a single guide RNA. In addition, Cas12i1 was reported to have pre-crRNA processing and collateral ssDNA cleavage activities. However, the detailed molecular mechanisms underlying the Cas12i function remain ill-defined, and no structural information is currently available. Furthermore, whether Cas12i displays other cleavage activities remains unknown.

Here, we have determined the crystal structures of Cas12i2 in complex with crRNA and PAM-containing target DNA, as well as a Cas12i2–crRNA binary complex. The DNA target strands used for crystallization contain spacers of varying lengths (12- and 26-nt) to capture different conformations of Cas12i2. In the DNA-bound Cas12i2 ternary complex, a 5-nt single-stranded DNA (ssDNA) and two $Mg^{2+}$ ions are observed in the RuvC catalytic pocket, showing how the catalytic residues, together with the DNA substrate, coordinate metal ions in the RuvC-containing Cas proteins including Cas12 and Cas9. In addition to the double-strand DNA (dsDNA) cleavage activity, Cas12i2 cleaves ssDNA in *trans* upon crRNA forming heteroduplex with the complementary ssDNA, as well as processes the pre-crRNA to

form the mature crRNA. Together, our structural and biochemical characterization of Cas12i2 provides insights into the molecular mechanisms of this CRISPR effector and suggests potential applications of Cas12i2.

## Results

**The Cas12i2–crRNA–dsDNA ternary complex.** To investigate the structure and mechanism of Cas12i2–crRNA loading and target DNA recognition, we determined the crystal structure at 2.57 Å resolution of wild-type (wt) Cas12i2 in complex with a 54 nucleotide (nt) crRNA, comprises 23-nt repeat-derived segment (hereinafter referred to as repeat) and a 31-nt spacer-derived segment (hereinafter referred to as spacer), and a partial dsDNA containing a 26-nt complementary target strand and a 5′-TTC-3′ protospacer adjacent motif (PAM) (PDB ID: 6LTU) (Fig. 1a, b and Supplementary Fig. 1a), as well as the crystal structure of the catalytic inactive Cas12i2$^{E833A}$ mutant in complex with the same crRNA and dsDNA at 2.5 Å resolution (PDB ID: 6LTR) (Supplementary Fig. 1b and Table 1). The overall structures of these two ternary complexes are identical. Our structural homology searches using the Dali server show that the overall structure of Cas12i2 exhibits distant similarity with AacCas12b, in agreement with their distant sequence similarity.

The Cas12i2 consists of the recognition (REC) and nuclease (NUC) lobes connected by the wedge (WED) domain (Fig. 1c). The REC lobe is composed of the Helical-I, Helical-II, and PAM-interacting (PI) domains. The NUC lobe consists of the Helical-III, WED, Bridge helix (BH), RuvC, and Nuc domains. The Helical-I and Helical-II domains form a dumbbell-like structure, with the domains located at the PAM-proximal and PAM-distal region of the guide:target heteroduplex, respectively. The Helical-III domain interacts with the PAM-distal region of the guide:target heteroduplex from the opposite side to the Helical-II domain. The RuvC and Nuc domains are positioned side-by-side, forming a positively charged concavity, in which an ssDNA is bound. One side of the WED domain contacts the 23-nt crRNA repeat and the opposite site interacts with the dsDNA (Fig. 1c). Most nucleotides of the crRNA (49 out of 54) and all DNA nucleotides are visible (Supplementary Fig. 1c, d). The crRNA spacer region is paired with the complementary target DNA, and the crRNA:DNA heteroduplex is located in the cleft formed by the REC and NUC lobes.

**Recognition of the crRNA repeat.** The repeat region (nucleotide G(−1)-A(−23)) of the 54-nt crRNA forms a stem-loop carrying a 6-nt 5′-tail, and is positioned in the positively charged cleft formed by the WED, BH, and Helical-III domains (Supplementary Fig. 2a). The stem-loop consists of a 6-bp stem, a 2-nt bulge and a 3-nt loop. The repeat region forms extensive hydrogen bonds with the WED, BH and Helical-III domains via its sugar-phosphate backbone and bases (Fig. 1d). For example, Lys537 forms hydrogen bonds with G(−15) and U(−14) from the major groove, and Gln809, Gln648, and Glu640 contact bases within the stem from the minor groove (Fig. 1e). Cas12i2 also interacts sequence specifically with the bases within the bulge, the loop, and the 5′-tail of the repeat (Fig. 1f–h and Supplementary Fig. 2b). Base contacts between the repeat and Cas12i2 are more numerous than seen in other crRNA-bound Cas proteins, including Cas9, Cas12a, Cas12b, and Cas13a. Point mutation of residue that contacts the repeat had slight effect on DNA cleavage activity (Supplementary Fig. 2c). In addition, one $Mg^{2+}$ ion binds in the cleft formed by the loop, thus stabilizing the bases of C(−9) and U(−12) via water molecules.

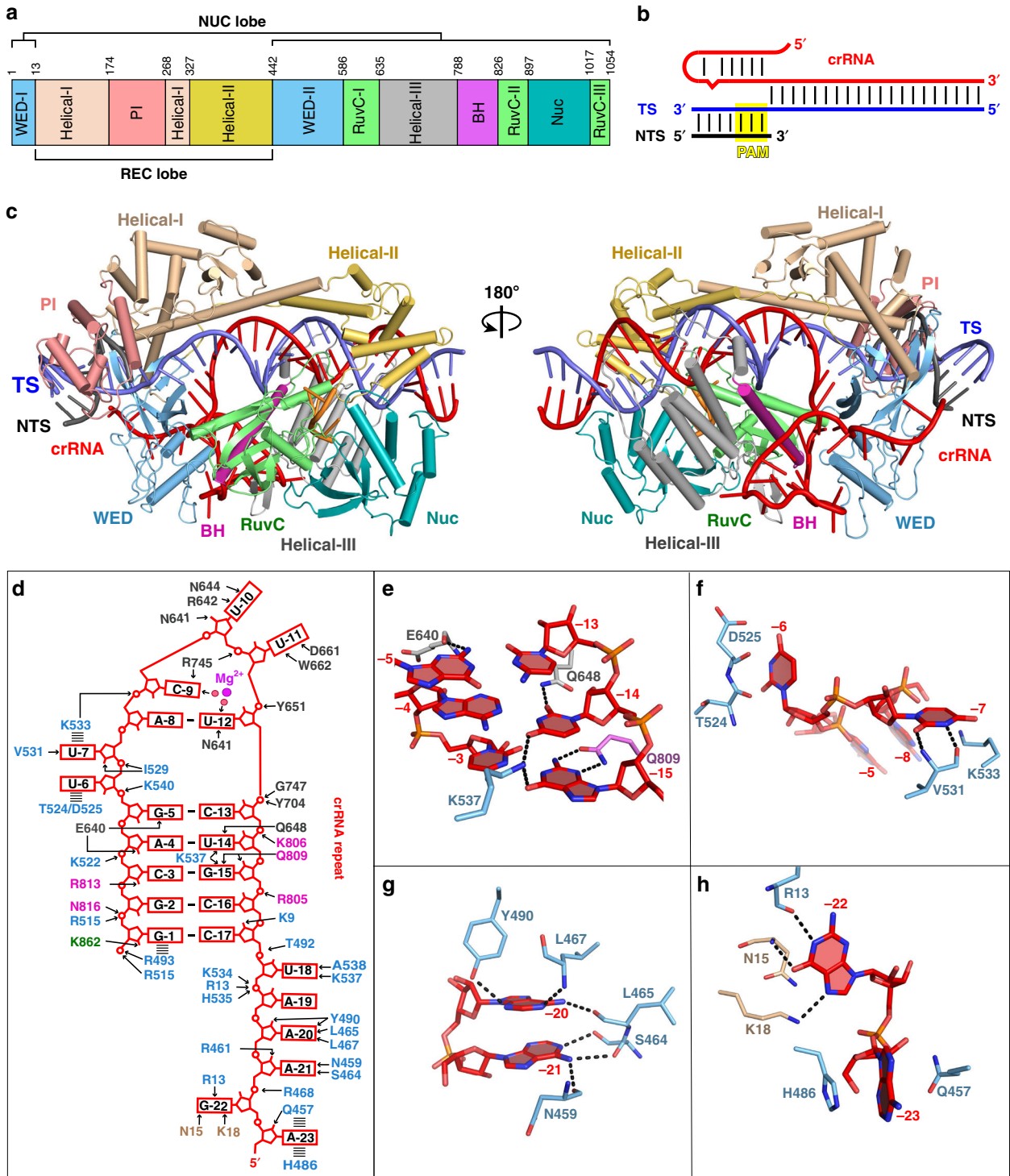

**Fig. 1 Overall structure of Cas12i2−crRNA−DNA ternary complex and The crRNA recognition. a** Cas12i2 domain organization. **b** Schematic representation of the crRNA and dsDNA used for crystallization. crRNA, target strand DNA, and non-target strand DNA are shown in red, blue, and black, respectively. The crRNA repeat region is the hairpin at the 5'-end. **c** Two views of ribbon presentation of the Cas12i2wt-crRNA-dsDNA ternary complex (PDB ID: 6LTU). Domains of Cas12i2, crRNA, and dsDNA are colored according to the scheme used in **a** and **b**. The single-stranded DNA bound in the RuvC domain is shown in orange. **d** Contacts between Cas12i2 residues and the hairpin repeat region of the crRNA. Hydrogen bond and salt bridge interactions are indicated with black arrows, and the stacking interactions are shown by dashed lines. **e** Magnified view of the base-contacts between Cas12i2 and the crRNA stem. Hydrogen bonds between Cas12i2 and the repeat are shown as dashed lines. **f** Interactions between the WED domain and two flipped out nucleotides. **g**, **h** Magnified views of the interactions between Cas12i2 and **g** nucleotides A(−21) and A(−20), and **h** nucleotides A(−23) and G(−22) within the 5'-handle of the crRNA.

**Table 1 Data collection and refinement statistics.**

| | Se-Cas12i2$^{E833A}$-crRNA-dsDNA (26 nt) | Cas12i2$^{WT}$-crRNA-dsDNA (26 nt) | Cas12i2$^{E833A}$-crRNA-dsDNA (26 nt) | Cas12i2$^{E833A}$-crRNA-dsDNA (12 nt) | Cas12i2$^{WT}$-crRNA |
|---|---|---|---|---|---|
| PDB code | | 6LTU | 6LTR | 6LU0 | 6LTP |
| **Data collection** | | | | | |
| Beamline | SSRF BL19U1 | SSRF BL17U1 | Spring-8 BL41XU | SSRF BL17U1 | SSRF BL19U1 |
| Space group | C222$_1$ | C222$_1$ | C222$_1$ | C222$_1$ | P4$_3$ |
| Cell dimensions | | | | | |
| $a$, $b$, $c$ (Å) | 94.7, 123.8, 284.5 | 94.3, 123.6, 281.2 | 93.3, 123.0, 280.4 | 93.6, 122.7, 284.7 | 146.3, 146.3, 144.6 |
| $\alpha$, $\beta$, $\gamma$ (°) | 90.0, 90.0, 90.0 | 90.0, 90.0, 90.0 | 90.0, 90.0, 90.0 | 90.0, 90.0, 90.0 | 90.0, 90.0, 90.0 |
| Wavelength (Å) | 0.981 | 0.979 | 1.000 | 0.979 | 0.979 |
| Resolution (Å) | 50.00–2.90 (2.97–2.90)$^a$ | 50.00–2.57 (2.61–2.57) | 50.00–2.50 (2.54–2.50) | 50.00–3.20 (3.26–3.20) | 50.00–3.40 (3.46–3.40) |
| $R_{merge}$ (%) | 14.9 (116.6) | 16.0 (98.5) | 5.4 (35.3) | 9.3 (86.2) | 13.2 (68.4) |
| $R_{pim}$ (%) | 4.0 (30.4) | 6.6 (39.5) | 2.2 (16.5) | 3.9 (36.8) | 5.5 (32.3) |
| $I/\sigma I$ | 18.3 (2.4) | 12.7 (2.2) | 31.6 (4.2) | 26.5 (4.0) | 13.3 (2.4) |
| Completeness (%) | 100.00 (100.00) | 99.30 (100.00) | 99.00 (94.80) | 99.80 (99.50) | 100.00 (100.00) |
| Redundancy | 14.8 (15.5) | 6.8 (7.2) | 6.6 (5.1) | 6.4 (6.2) | 6.6 (5.4) |
| **Refinement** | | | | | |
| Resolution (Å) | | 46.87–2.57 | 46.74–2.50 | 29.56–3.20 | 48.71–3.40 |
| No. of reflections | | 51,108 | 54,751 | 26,260 | 39,683 |
| $R_{work}$ / $R_{free}$ (%) | | 20.20/22.88 | 19.75/22.63 | 20.69/23.06 | 20.74/23.02 |
| Number of atoms | | | | | |
| Protein | | 8313 | 8327 | 7448 | 14,592 |
| Nucleic acid | | 2060 | 2076 | 1250 | 1778 |
| Ligand | | 7 | 5 | 0 | 0 |
| Water | | 325 | 392 | 21 | 12 |
| B-factors (Å$^2$) | | | | | |
| Protein | | 49.58 | 40.06 | 63.89 | 48.91 |
| Nucleic acid | | 54.21 | 43.40 | 45.18 | 71.65 |
| Ligand | | 39.34 | 22.92 | | |
| Water | | 39.03 | 33.35 | 33.30 | 15.47 |
| R.m.s. deviations | | | | | |
| Bond lengths (Å) | | 0.007 | 0.008 | 0.004 | 0.006 |
| Bond angles (°) | | 1.062 | 1.165 | 0.794 | 1.153 |
| Ramachandran plot | | | | | |
| Favored (%) | | 97.77 | 96.71 | 94.83 | 95.29 |
| Allowed (%) | | 2.23 | 3.00 | 4.63 | 4.43 |
| Outliers (%) | | 0 | 0.29 | 0.54 | 0.27 |

$^a$Numbers in parentheses represent statistics in highest resolution shell.

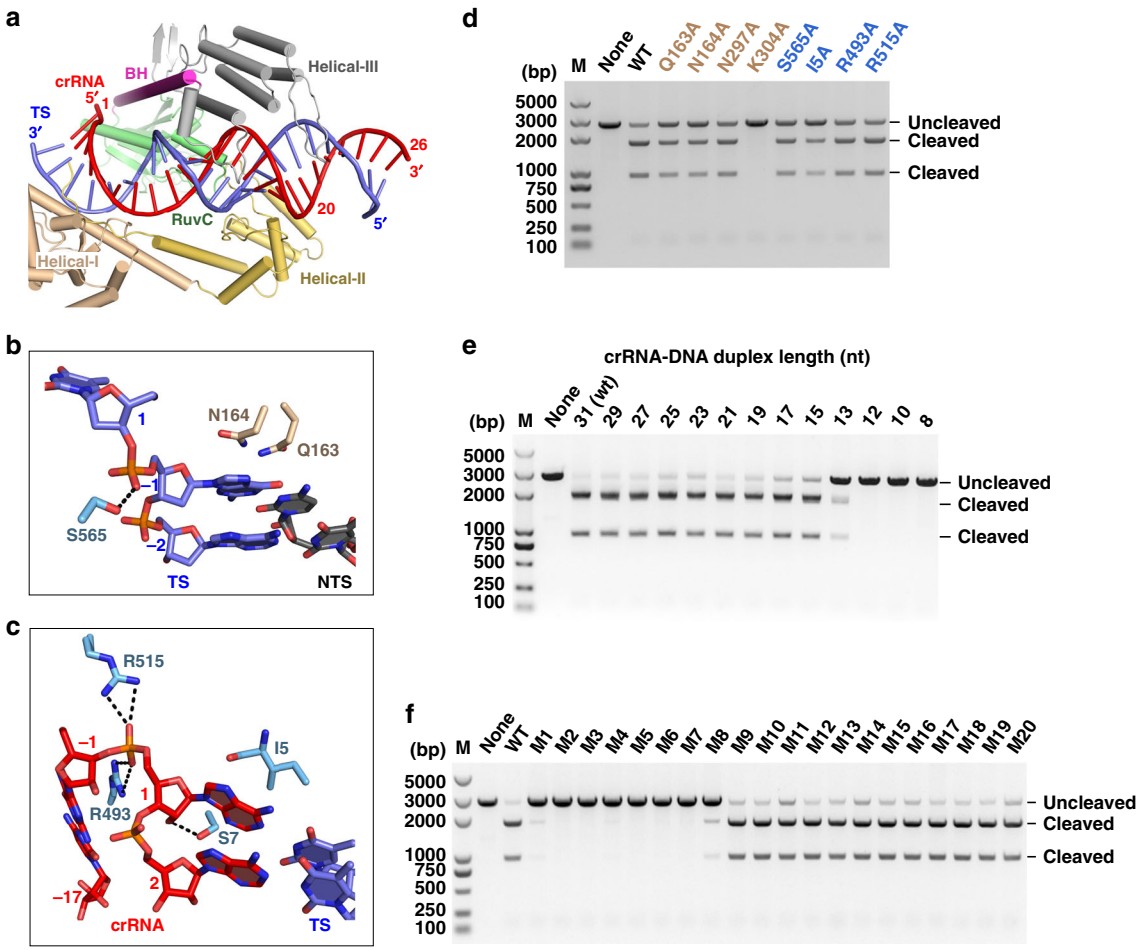

**Fig. 2 The guide:target heteroduplex recognition. a** The guide:target heteroduplex is located in the cleft between the REC and NUC lobes. **b** Magnified view of interactions between Cas12i2 and nucleotide (−1) of the target dsDNA. **c** Interaction of the WED domain with the RNA:DNA hybride at position +1. **d** Mutation of residues interacting with dsDNA or RNA:DNA duplex at position 1–2 impairs the DNA cleavage activity by Cas12i2. Linear plasmid DNA cleavage assay using the 300 ng linearized plasmid DNA as substrate in the presence of 200 nM mutants or wild-type Cas12i2–crRNA complex at 37 °C for 15 min. **e** Linear plasmid DNA cleavage assay showing that the length of the guide:target heteroduplex affects the cleavage activity of Cas12i2. **f** Linear plasmid DNA cleavage assay using single mismatched target DNA.

**Target search and recognition**. In the DNA-bound Cas12i2 ternary complex, the crRNA spacer region and the complementary target strand form a 26-bp crRNA:DNA heteroduplex. This heteroduplex lies in the central channel formed by REC and NUC lobes and is stabilized by the hydrogen-bonding and ionic interactions between backbone sugar-phosphate groups and the Helical-I–III, RuvC, and WED domains of Cas12i2 (Supplementary Fig. 3). Base pairs 1–19 in the PAM-proximal region interact via sugar-phosphate backbone contacts primarily with the Helical-I–III and RuvC domains, while base pairs 20–26 in the PAM-distal region protrude from the Cas12i2 protein (Fig. 2a). The stacking interactions between dsDNA base (−1) and residues Asn164 and Gln163 prevent further base-pairing within the dsDNA beyond position (−1), thus facilitating the unwinding of the dsDNA and the base-pairing between the crRNA and the target (Fig. 2b). Furthermore, the stacking interactions between the crRNA nucleotides A1 and G(−1) and residues Ile5 and Arg493 further stabilize the crRNA:DNA heteroduplex and the formation of the R-loop structure (Fig. 2c). Notably, the alanine replacement of these residues to remove the interactions between Cas12i2 and crRNA or target DNA reduced the activity of Cas12i2, and the Lys304Ala mutation almost completely abolished DNA cleavage (Fig. 2d).

To test whether the length of the guide:target heteroduplex affects the activity of Cas12i2, we performed DNA cleavage assays using linear plasmid DNAs containing complementary targets of various lengths (Fig. 2e). Target DNA containing >15 nucleotides complementary to the guide are cleaved at the similar level as WT, whereas 13-nt of complementarity abolished cleavage, indicating a >13-bp guide:target heteroduplex is required for Cas12i2 dsDNA cleavage activity. We tested the effect of a single mismatch between the guide and target on dsDNA cleavage activity. Single mismatches between the guide and nucleotides 1–8 of the crRNA abrogated activity, whereas DNA cleavage was unaffected by single mismatches in nucleotides 9–20 (Fig. 2f), suggesting that base-pairing between the guide and target at the PAM-proximal region is essential. Nucleotides 1–8 of the crRNA are thus likely to present the "seed" region.

**PAM sequence recognition**. The PAM duplex is located in the cleft formed by the WED and PI domains (Fig. 3a), and is stabilized by interactions between the PI domain and the phosphate backbone (Fig. 3b). The Ala232 main-chain forms a hydrogen bond with the base of nucleotide T(−2′) on the non-target DNA strand. This hydrogen bond is the sole direct base contact between Cas12i2 and the PAM duplex, which is distinct from

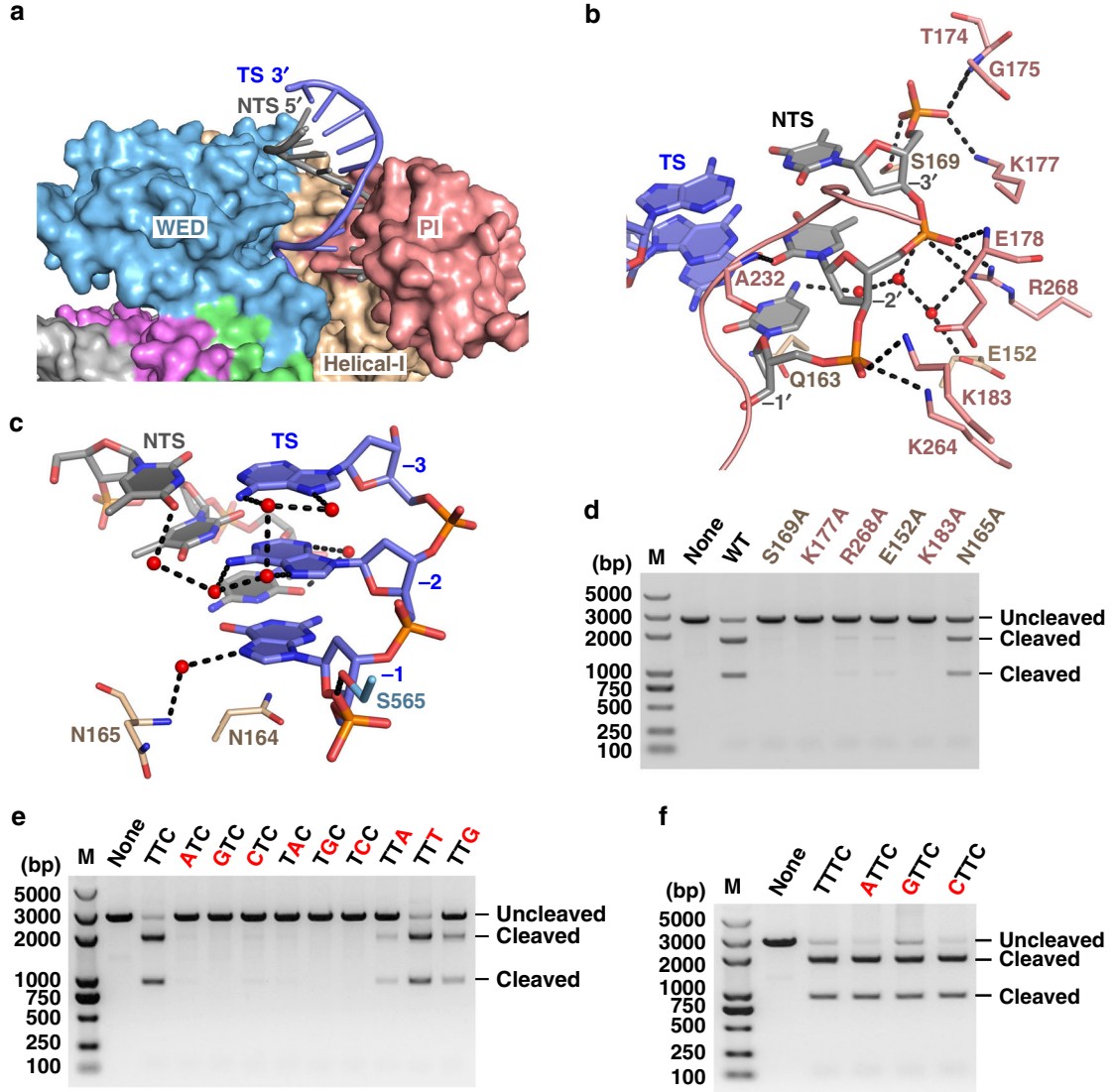

**Fig. 3 The PAM duplex recognition. a** Recognition of the PAM duplex by the WED and PI domains. **b** The PI domain recognizes the base of T(−2′) via Ala232, and forms extensive hydrogen bonds with the phosphate backbone of the non-target strand. **c** Most bases of the PAM duplex form extensive hydrogen bonds with either Cas12i2 or other bases within the PAM duplex via waters (red spheres), forming a water-mediated hydrogen bond network. **d** Mutation of residues forming hydrogen bonds with the PAM sequence substantially reduces the DNA cleavage. **e** DNA cleavage by Cas12i2 using a linear plasmid DNA containing a mutant PAM sequence. The mutated nucleotide is shown in red and the wild-type sequence is shown in black. **f** The mutation of the nucleotide T that is directly upstream the PAM sequence shows no effect on Cas12i2 DNA cleavage activity.

PAM recognition by Cas12a, Cas12b, or Cas9, which form multiple direct hydrogen bonds with PAM bases[9,16,25,26]. In contrast, in the DNA-bound Cas12i2 ternary complex, the bases of the PAM duplex form extensive indirect hydrogen bonds via water molecules with either Cas12i2 or other bases within the PAM duplex (Fig. 3c).

To test the significance of this water-mediated hydrogen bond network, we assayed the effects of several mutations of either Cas12i2 or PAM sequence on the DNA cleavage activity. The DNA cleavage were severely impaired or completely abrogated by mutation of residues that interact with the PAM sequence (Fig. 3d). Furthermore, replacement of T(−2′) and T(−3′) by any other nucleotide abolished DNA cleavage, while substitution of C(−1′) by nucleotide A or G substantially reduced DNA cleavage (Fig. 3e). In contrast, mutation of adjacent nucleotides has no effect on DNA cleavage (Fig. 3f). Thus, the T(−2′) and T(−3′) bases are required for cleavage, and interactions between the PAM duplex and Cas12i2 have

essential roles in the recognition of the PAM sequence during the DNA target search.

**ssDNA binding in the RuvC catalytic pocket**. The RuvC domain is the sole DNase catalytic domain in Cas12i2, with catalytic residues Asp599, Glu833, and Asp1019 lining the active pocket. In the DNA-bound Cas12i2 ternary complexes, a 5-nt ssDNA, which was probably introduced with the target DNA, is observed in the RuvC catalytic pocket (Fig. 4a and Supplementary Fig. 4a). In addition, the ssDNA bound in the catalytic pocket remains intact may due to the high concentration of the complex and the chemicals used for crystallization, suggesting that our DNA-bound Cas12i2 complex is in the substrate binding state. The phosphate group between the 2nd and the 3rd nucleotides is located in the catalytic pocket, indicating that this is the cleavage position. The Phe392 side-chain inserts between nucleotides 3 and 4 of the ssDNA

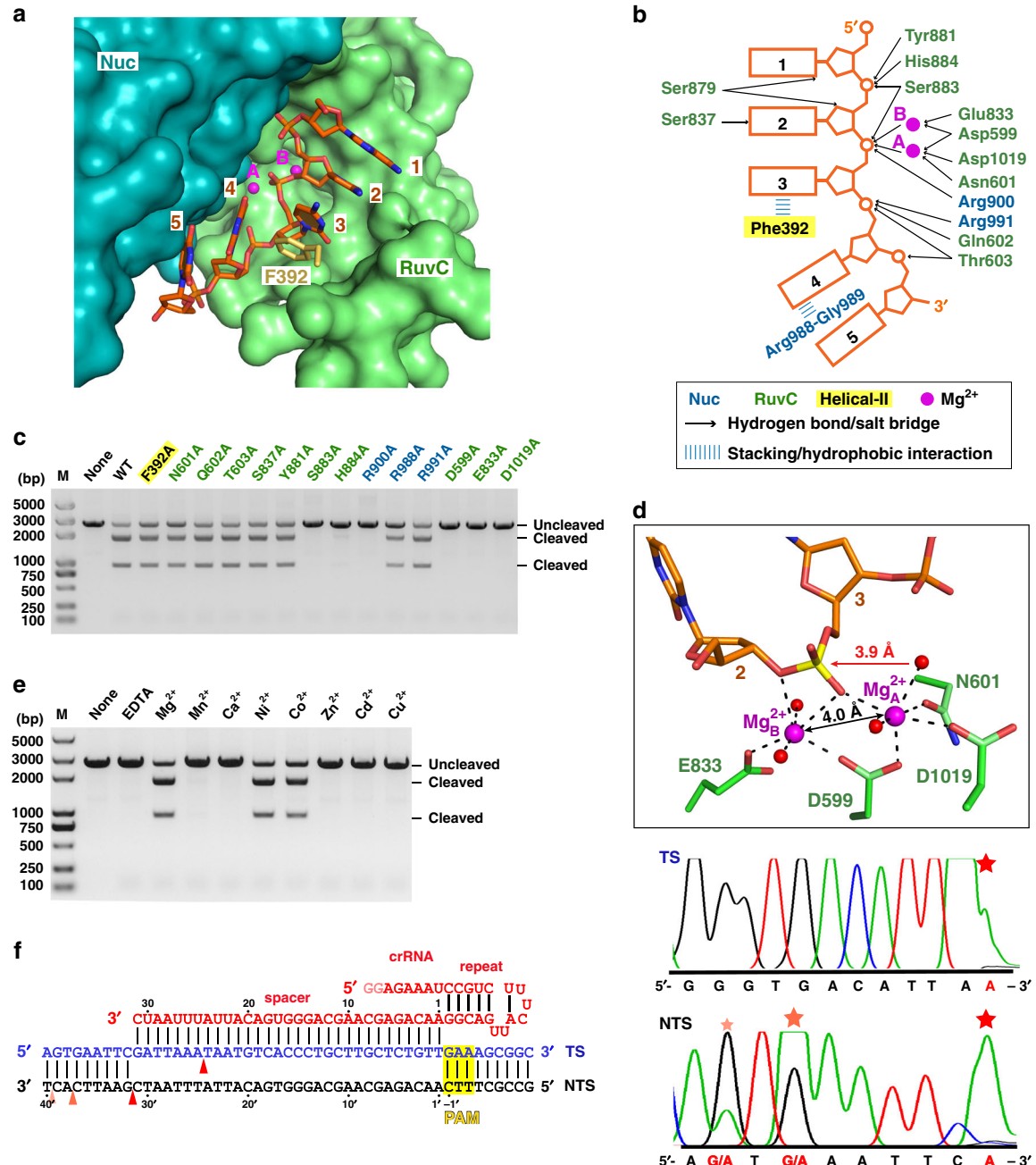

**Fig. 4 ssDNA substrate bound in the RuvC catalytic pocket in the Cas12i2$^{wt}$–crRNA–DNA ternary complex. a** 5-mer substrate ssDNA is positioned in the cleft formed by the RuvC and Nuc domains. The nucleotide sequence of the ssDNA in the catalytic pocket is assumed based on its electron density. **b** Schematic interactions of substrate DNA and the RuvC and Nuc domains. Hydrogen bonds and salt bridges are indicated with black arrows. Stacking interactions are shown by blue lines. Two Mg$^{2+}$ ions are shown in magenta spheres. **c** Mutational analysis of key residues interacting with ssDNA or Mg$^{2+}$. **d** Magnified view of the coordination of two Mg$^{2+}$ ions with RuvC catalytic residues, the scissile phosphate, and waters. The Mg$^{2+}$ ions are shown as large magenta spheres, and water molecules as small red spheres. The ssDNA substrate is shown in orange, the scissile phosphate in yellow. The distance between the phosphorus atom and nucleophile water is indicated, as well as the distance between two Mg$^{2+}$ ions. **e** Linear plasmid DNA cleavage assay using different metals. **f** Sanger-sequencing traces from Cas12i2-cleaved targets show staggered overhangs. The major and minor cleavage sites are indicated by red and orange triangles, respectively (left panel). The non-templated addition of an additional adenine, indicated by a red star, is an artifact of the polymerase used for sequencing (right panel).

substrate, resulting in a kink between nucleotides 3 and 4 (Fig. 4b). The ssDNA substrate is further stabilized by the hydrogen bonds and ionic interactions between backbone phosphate groups and side chains of His884, Ser883, Tyr881, Gln602, and Thr603 of the RuvC domain, as well as the side chains of Nuc domain residues Arg900 and Arg991. In addition, alanine substitution of Ser883, His884, and Arg900, which

form hydrogen bonds with the phosphate group of nucleotides 2 or 3, abolished Cas12i2 DNA cleavage activity (Fig. 4c), indicating that the stabilization of the nucleotides 2 and 3 is essential for DNA cleavage by RuvC domain. Our findings confirm that the RuvC domain harbors the catalytic pocket, with the RuvC and Nuc domains being crucial for ssDNA substrate binding.

**$Mg^{2+}$ cations mediate ssDNA cleavage activity**. In the Cas12i2$^{wt}$–crRNA–DNA ternary complex, a pair of $Mg^{2+}$ cations (A and B), separated by 4.0 Å, are located on either side of the scissile phosphate between nucleotides 2 and 3 (Fig. 4d and Supplementary Fig. 4b). Both metal ions coordinate the scissile phosphate group and RuvC domain catalytic residues. $Mg^{2+}_A$ coordinates the side chains of Asp599, Asp1019, and Asn601, while $Mg^{2+}_B$ coordinates with the side chains of Glu833 and Asp599. In addition to catalytic residues and the scissile phosphate, each $Mg^{2+}$ also coordinates two water molecules. One water coordinating $Mg^{2+}_A$ is positioned 3.9 Å away from the scissile phosphate, and is located approximately in line with the scissile phosphate and the leaving oxygen atom, suggesting this water acts as the nucleophile that attacks the phosphorus of the scissile phosphate. $Mg^{2+}_B$ coordinates the 3′ oxygen atom leaving group, indicating the cleavage products contain 5′-phosphate and 3′-OH groups. Thus the two Mg ions facilitate DNA hydrolysis with $Mg^{2+}_A$ coordinating and activating the water nucleophile, and $Mg^{2+}_B$ stabilizes the leaving group.

In the Cas12i2$^{E833A}$ ternary complex, $Mg^{2+}_A$ is seen, but $Mg^{2+}_B$ is missing due to the Glu833 mutation. The $Mg^{2+}_A$ cation is located in a similar position to $Mg^{2+}_A$ in the Cas12i2$^{wt}$ complex. However, unlike the Cas12i2$^{wt}$ $Mg^{2+}_A$, the Cas12i2$^{E833A}$ $Mg^{2+}_A$ coordinates an extra water and not the scissile phosphate (Supplementary Fig. 4c). In the Cas12i2$^{E833A}$ complex, the nucleophile water is no longer positioned in-line with the phosphate and the leaving group. Superposition of the Cas12i2$^{wt}$ and Cas12i2$^{E833A}$ complexes shows that the scissile phosphate in the Cas12i2$^{E833A}$ complex shifts ~1.3 Å away from $Mg^{2+}_A$ (Supplementary Fig. 4d), preventing their coordination. Thus, the binding of two $Mg^{2+}$ ions is essential for the proper conformation of ssDNA substrate. A similar structural arrangement of the DNA substrate is seen in the *Thermus thermophilus* Argonaute (TtAgo)[27,28].

Alanine substitution of any of the three catalytic residues or Asn601 impairs coordination of two $Mg^{2+}$ ions and completely abolished DNA cleavage for the catalytic residues, and reduces DNA cleavage for Asn601 (Fig. 4c). We tested the cations specificity required for DNA cleavage and found that $Mg^{2+}$, $Ni^{2+}$, and $Co^{2+}$ facilitate dsDNA cleavage by Cas12i2, whereas $Mn^{2+}$, $Ca^{2+}$, $Zn^{2+}$, $Cd^{2+}$, or $Cu^{2+}$ ions do not facilitate cleavage (Fig. 4e). Thus, divalent cations are essential for DNA cleavage by the RuvC domain.

To identify the dsDNA cleavage site, we performed endonuclease assays using the linear plasmid DNA together with Sanger sequencing, and found that Cas12i2 cleaved the target strand between nucleotides 24 and 25 (Fig. 4f). The non-target strand was cleaved at three sites, with the major cleavage site between nucleotides 31–32, and minor cleavage sites between nucleotides 37–38 or 39–40. Thus, Cas12i2 cleaves the plasmid DNA on the PAM-distal site, resulting in 7-nt, 13-nt or 15-nt overhangs. This is similar to Cas12a and Cas12b, which also cleave the target dsDNA at the PAM-distal site generating the product with staggered end[9,13,15].

**The Cas12i2–crRNA binary complex**. To explore how the Cas12i2–crRNA binary complex assembles, we determined its crystal structure at 3.4 Å resolution (PDB ID: 6LTP) (Fig. 5a, Supplementary Fig. 5a and Table 1). In contrast to the DNA-bound Cas12i2 ternary complex, in which 26 nucleotides within the crRNA spacer are traceable, the spacer region is mostly disordered in the binary complex. Only 4 nucleotides in the seed region, 6 nucleotides in the central area and 9 nucleotides at the 3′-ends of the spacer were observed in the binary complex, suggesting the crRNA seed region is only partially pre-ordered. In

contrast, the seed regions of Cas9 and Cas12b are fully pre-ordered[9,15,16,29]. In addition, the PI domain is almost completely disordered in the Cas12i2–crRNA binary complex, while it is ordered in all DNA-bound Cas12i2 ternary complexes, suggesting that the PI domain is not pre-ordered prior to DNA binding and is stabilized by interactions with the PAM sequence (Supplementary Fig. 5b). By contrast, the PI domain is ordered in the guide RNA-bound Cas12a, Cas12b, or Cas9[9,12,16,30].

**Cas12i2–crRNA–DNA target recognition state structure**. To further clarify how Cas12i2 recognizes the target DNA via the crRNA seed region, we determined the crystal structure of Cas12i2 in complex with the crRNA and the partial dsDNA comprising a 12-nt protospacer at 3.2 Å resolution (PDB ID: 6LU0) (Fig. 5b and Supplementary Fig. 5c). In this ternary complexes, we observed an 8-bp guide:target heteroduplex. Given that the target only base pairs with the crRNA seed region, this complexe likely represent the nucleation step of the guide:target heteroduplex formation. The Helical-II domain is mostly disordered in this structure, whereas the other domains have conformations similar to those in the fully base-pairing DNA-bound Cas12i2 ternary complex, suggesting that base pairing between the guide and target induces the conformational change seen in the Helical-II domain.

In contrast to the fully complementary target-bound Cas12i2 ternary complex, where a short ssDNA is observed in the RuvC catalytic pocket, neither ssDNA nor a $Mg^{2+}$ ion was present in the Cas12i2 RuvC domain in the nucleation state. This observation suggests that in the 12-nt spacer DNA-bound ternary complex the RuvC catalytic pocket is in the inactive state, while it is in the active state in the fully paired DNA-bound ternary complex, revealing why the Cas12i2 in complex with a crRNA:DNA duplex of 12-bp or shorter lacks DNA cleavage activity, whereas a complex with a >13-bp duplex exhibits high activity (Fig. 2e).

**Helical-II domain conformational change activates the RuvC domain**. Comparison of the binary and ternary complexes shows that the Helical-II domain is ordered in both the Cas12i2–crRNA binary and the fully paired DNA-bound Cas12i2 ternary complexes, while it is flexible in the partially paired DNA-bound complexes (Fig. 5c and Supplementary Fig. 5d), suggesting that the base-pairing in the seed region increases the flexibly of the Helical-II domain. Structural superposition of the RuvC domains in the partial-paired and the fully paired DNA-bound ternary complexes shows that the RuvC catalytic pocket undergoes a conformational change upon substrate DNA binding (Fig. 5d).

In the Cas12i2–crRNA binary complex, the Helical-II domain that is positioned closed to the RuvC domain acts like a lid covering and closing off the RuvC catalytic pocket (Fig. 5e). In addition, the loop (residues 386–393) within the Helical-II domain inserts into the RuvC catalytic pocket, occupying the substrate ssDNA binding position. Therefore, the Helical-II domain blocks the active site of the RuvC domain prior to the target DNA binding. In contrast, the Helical-II domain rotates away from the other Cas12i2 domains upon target strand binding. The outward rotation of the Helical-II domain in the fully paired DNA-bound Cas12i2 ternary complex opens the RuvC catalytic pocket, rendering it accessible to the ssDNA substrate (Fig. 5f). Thus, RuvC catalytic pocket is inactive prior to formation of the guide:target heteroduplex, which induces a rearrangement of the Helical-II domain and activates the RuvC catalytic pocket.

**Cas12i2 cleaves ssDNA non-sequence specifically**. To test whether target ssDNA triggers Cas12i2 *trans* DNA cleavage

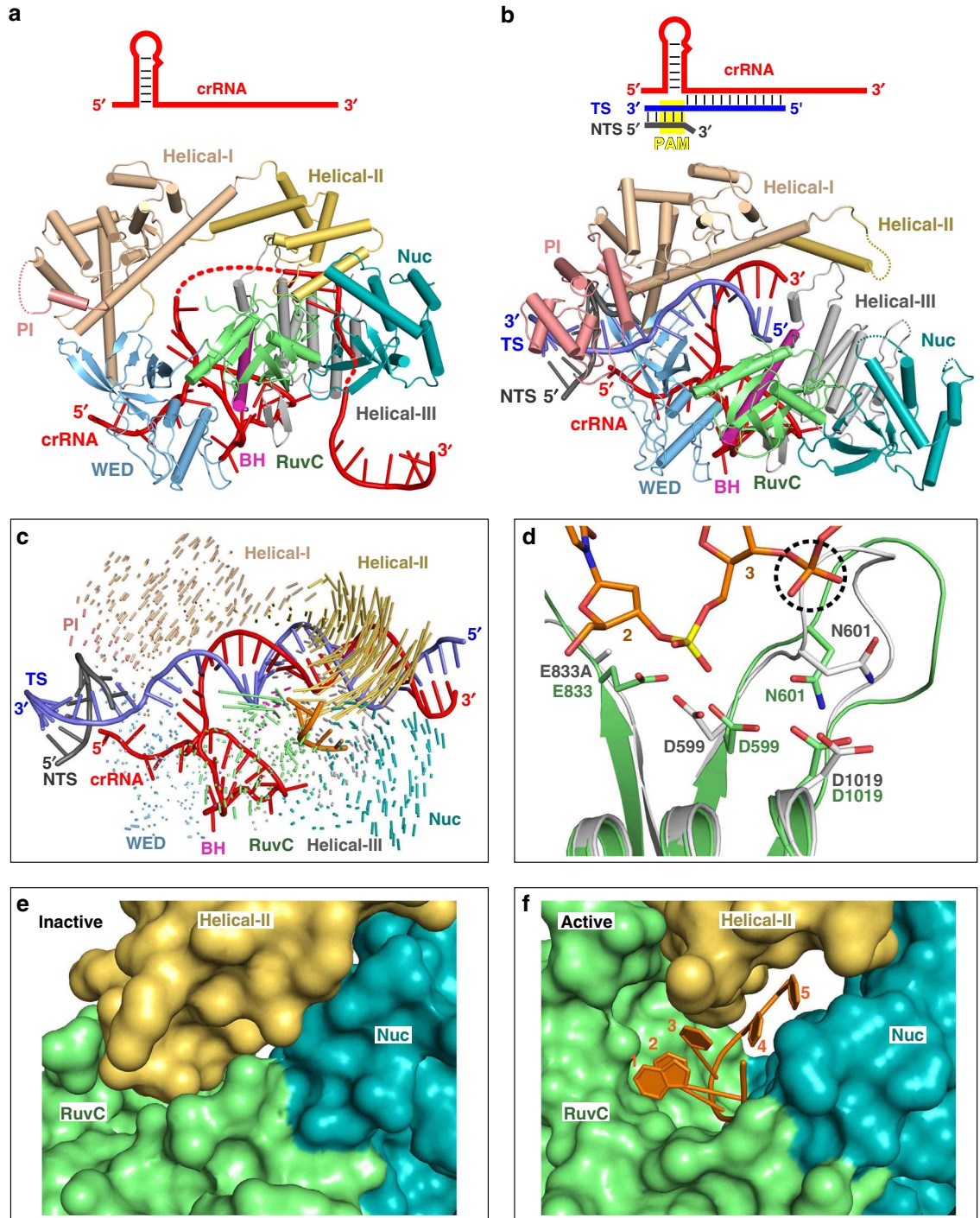

**Fig. 5 Conformational rearrangement of the Helical-II domain activates the RuvC domain. a** Crystal structures of the Cas12i2–crRNA binary complex (PDB ID: 6LTP). **b** Crystal structures of the 12-nt DNA-bound Cas12i2 ternary complex, which represents nucleation of the guide and target strands (PDB ID: 6LU0). **c** Superposition of Cas12i2–crRNA binary complex and fully paired DNA-bound ternary complex showing the conformational change upon the guide:target heteroduplex forming. **d** Structural comparison of RuvC catalytic pocket of 12-nt (in gray) and 26-nt (green) DNA-bound Cas12i2 ternary complexes. The phosphate group of the 4th nucleotide in the ssDNA substrate, which clashes with a loop in the partial-paired complex, is highlighted by black dashed circle. **e**, **f** Structural comparison of the RuvC, Nuc, and Helical-II domains in the RuvC inactive (**e**) and active (**f**) states, showing that the structural rearrangement of the Helical-II domain opens the RuvC catalytic pocket.

activity, we performed cleavage assays using 5′-Cy3-labeled ssDNA as the substrate, together with various DNA molecules as activators (Fig. 6a). Cas12i2 DNase activity was initiated on ssDNA complementary to the crRNA with or without the flanking PAM, as well as the unpaired dsDNA within the protospacer activated low levels of DNase activity. In contrast, non-complementary ssDNA and fully duplexed dsDNA that is complementary to the crRNA failed to activate the ssDNA cleavage of Cas12i2. Furthermore, in the presence of a complementary ssDNA activator, only ssDNA was cleaved (Fig. 6a), whereas dsDNA failed to be cleaved (Supplementary Fig. 6a). These data suggest that crRNA:DNA heteroduplex formation activates the

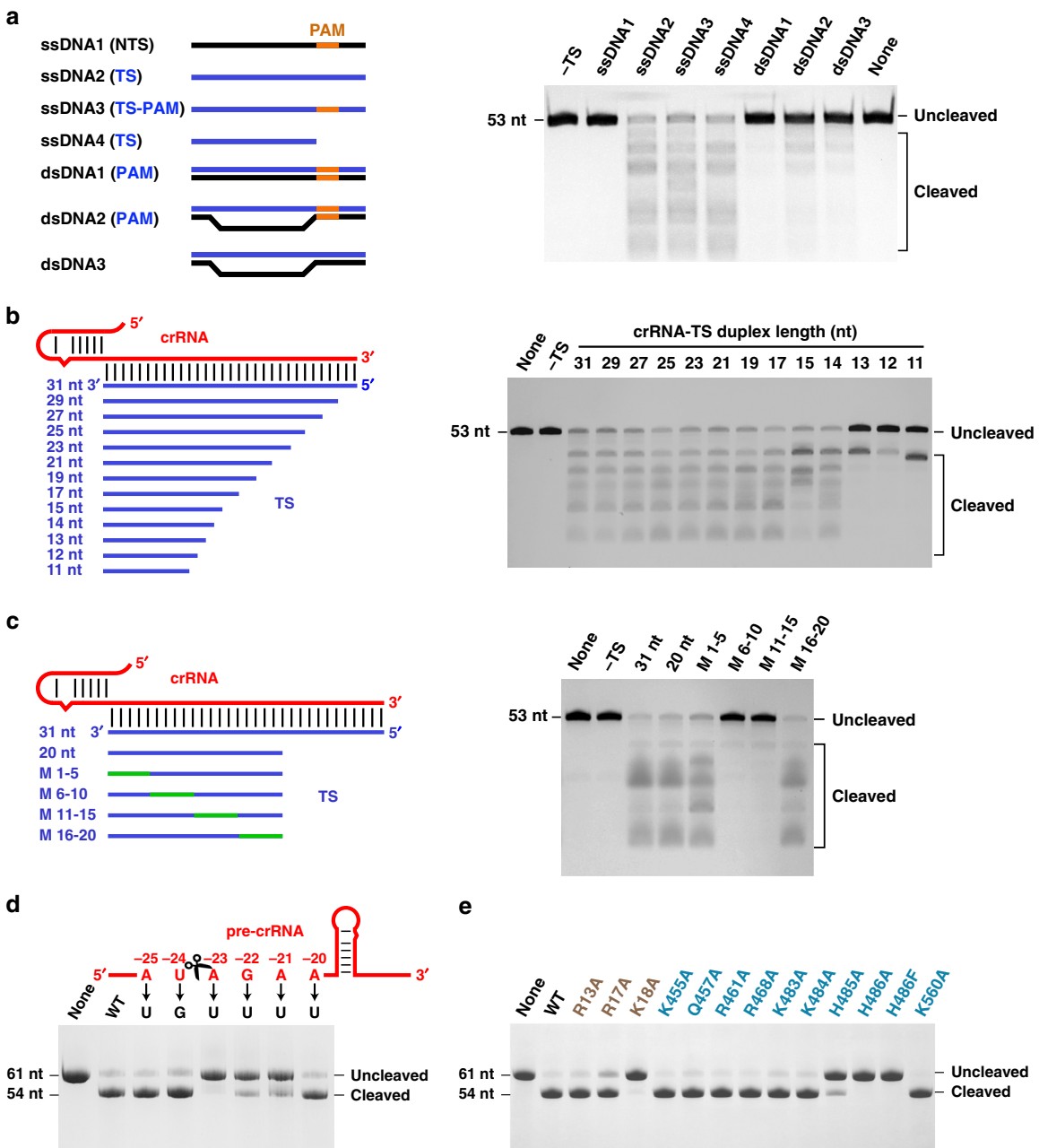

**Fig. 6 Cas12i2 target recognition activates non-specific ssDNA cleavage and pre-crRNA processing by Cas12i2. a** The formation of the guide:target heteroduplex activates the RuvC catalytic pocket, inducing the *trans* cleavage of ssDNA in a PAM-independent manner. (Left) Schematic representation of ssDNA and dsDNA activators. The complementary target DNA and non-complementary DNA are in blue and black, respectively. The PAM sequence is highlighted in orange. (Right) The DNA cleavage assay using the 200 nM 5'-Cy3-labeled ssDNA substrate in the presence of 200 nM DNA activators at 37 °C for 15 min. **b** The length of the complementary ssDNA activator affects non-specific DNA cleavage. (Left) Schematic representation of crRNA and ssDNA activators that is complementary to the guide RNA. (Right) The ssDNA cleavage assay in the presence of ssDNA activators with various lengths. **c** Mismatches between the target strand DNA and the crRNA spacer region affect activation of the RuvC catalytic pocket. (Left) Schematic representation of crRNA and ssDNA with mismatched regions are highlighted in green. (Right) The ssDNA cleavage assay in the presence of ssDNA activators containing mismatches. **d** Denaturing gel analysis showing that pre-crRNA processing by Cas12i2 depends on the crRNA repeat sequence shown in the up panel directly downstream of the cleavage site. The cleavage site is indicated by cartoon scissors. The pre-crRNA cleavage assay using the pre-crRNA mutants substrate in the presence of Cas12i2 at 37 °C for 30 min. **e** Pre-crRNA processing using Cas12i2 mutants.

Cas12i2 RuvC domain, triggering ssDNA cleavage in a non-sequence specific and PAM-independent manner.

The ssDNA cleavage assay in the presence of target DNAs of various lengths showed that 13-nt target DNA lacks the ability to initiate cleavage, while a 14-nt target activated non-specific

ssDNA cleavage (Fig. 6b), suggesting that the minimal length of the crRNA:target DNA duplex to activate the RuvC domain is 14-bp. These findings are in agreement with the length needed for dsDNA cleavage by Cas12i2. The comparison of ssDNA cleavage with either the complementary ssRNA or ssDNA activator shows

that both the complementary ssDNA and ssRNA are able to activate the RuvC catalytic pocket, though the efficiency of ssRNA is much lower than that of ssDNA (Supplementary Fig. 6b).

**The seed region is not essential for in *trans* ssDNA cleavage.** The base-pairing between the guide and target at the spacer seed region is essential for dsDNA cleavage activity of Cas12a, Cas12b and Cas9[7,8,15]. To test whether complementarity between the target strand and the crRNA seed region is crucial for ssDNA cleavage in *trans*, we performed DNA cleavage with ssDNA activators that contained a single mismatch at various positions within the spacer region. None of these single mismatches in the spacer affected ssDNA cleavage (Supplementary Fig. 6c), unlike single mismatches in the seed region, which abolished dsDNA cleavage (Fig. 2f). Mismatch of five contiguous nucleotides in the seed region (nucleotides 1–5) or nucleotides 16–20 failed to affect ssDNA cleavage, while mismatch of nucleotides 6–10 or 11–15 completely abolished ssDNA cleavage (Fig. 6c). These results suggest that base-pairing between the guide and the target strands in the seed region is not required for ssDNA cleavage in *trans*, whereas seed region complementarity is essential for dsDNA cleavage. In contrast, base-pairing of nucleotides 6–15, which are located in the cleft between the RuvC and Helical-II domains, is essential for ssDNA cleavage in *trans*. Thus, the seed region is not crucial for ssDNA binding, but is important in determining target dsDNA binding. Moreover, the base-pairing between the crRNA and target strand in the seed region is not necessary for activation of the RuvC domain, but is required for unwinding the target dsDNA and the formation of the crRNA:target heteroduplex.

**Cas12i2 processes pre-crRNA in a metal-independent manner.** To test whether Cas12i2 possesses pre-crRNA processing activity, we performed in vitro cleavage assays using a 61-nt pre-crRNA substrate in the presence of either divalent cations ($Mg^{2+}$, $Mn^{2+}$, or $Ca^{2+}$) or the chelating agent EDTA. The RNA substrate was cleaved in the presence of either divalent cations or EDTA (Supplementary Fig. 6d), yielding a shorter RNA containing a 23-nt repeat and 31-nt spacer. These results suggest that Cas12i2 is capable of processing its pre-crRNA in a metal-independent manner and that the cleavage site is located between the nucleotides A(−23) and U(−24).

To test whether the pre-crRNA sequence is critical for efficient pre-crRNA processing we generated pre-crRNAs containing point mutations at nucleotides (−25)-(−20) within the 5′-flanking region, and tested Cas12i2 processing efficiency (Fig. 6d). The single A(−23) mutation completely abolished Cas12i2 activity, and mutation of either nucleotides G(−22) or A(−21) substantially reduced activity. By contrast, mutation of other nucleotides showed no detectable effects on pre-crRNA processing. Thus, it is the sequence directly downstream the scissile site that is critical for pre-crRNA processing by Cas12i2, whereas the upstream sequence is not essential.

To identify the catalytic residues required for pre-crRNA processing, we screened Cas12i2 mutants for pre-crRNA cleavage activity (Fig. 6e). Alanine substitution of His486 or Lys18 completely abolished cleavage, and substitution of His485 substantially reduced cleavage activity, suggesting that residues His486, His485, and Lys18 are required for pre-crRNA processing. In addition, His486 and Lys18 contact the 5′-end of the mature crRNA (Fig. 1h), indicating that His486 and Lys18 act as the catalytic residues (Supplementary Fig. 6e). Furthermore, the crRNA processing is crucial for the DNA cleavage activity of Cas12i2. The mutation of amino acids Lys18, His485, or His486 decreases the DNA cleavage by Cas12i2–pre-crRNA complex, but

lacks effect on the Cas12i2–crRNA complex (Supplementary Fig. 6f).

## Discussion

Our structural analyses have revealed how Cas12i2 recognizes crRNA and target DNA. We show that the formation of the crRNA:DNA hybrid activates the RuvC catalytic pocket, which cleaves the dsDNA complementary to crRNA in *cis* and ssDNA in *trans*. In addition to its dsDNA and ssDNA cleavage activity, Cas12i2 possesses pre-crRNA processing activity for crRNA maturation.

In agreement with their distant sequence similarity (21.1%), Cas12i2 shares a distant similar overall structure with AacCas12b (Supplementary Fig. 7a, b). Superposition of the Helical-I–III domains of Cas12i2 and Cas12b shows that while their overall architectures are similar, the Helical-III domain failed to align well (Supplementary Fig. 7c). The comparison of the RuvC-Nuc of Cas12i2 and Cas12b showed that the core fold of their RuvC domains is similar, while their Nuc domains adopt different conformations (Supplementary Fig. 7d)[9,15]. Cas12i2, Cas12a, and Cas12b effectors possess RNA-guided dsDNA cleavage activities[7,21,24], resulting in a product containing staggered ends, probably ssDNA cleavage activity as well, suggesting that they exhibit both *cis*- and *trans* DNA cleavage activities. Importantly, the formation of the crRNA:DNA hybrid triggers the opening of the RuvC-Nuc cleft of Cas12i2 and Cas12a[14]. However, both maturation of guide RNA and the requirement for guide RNA for DNA cleavage differs for Cas12i2, Cas12a, and Cas12b. Cas12i2 and Cas12a possess an RNase activity responsible for crRNA maturation, while Cas12b lacks this activity. Cas12i2 and Cas12a only require crRNA for efficient DNA cleavage, whereas Cas12b requires both crRNA and tracrRNA.

Our biochemical studies have demonstrated that the seed region is essential for dsDNA cleavage, but not for ssDNA cleavage. The interactions between the PAM and Cas12i2 promote the melting of the target DNA duplex, enabling the target strand to base pair with the guide. For this reason, the PAM sequence and complementarity within the seed region are essential for dsDNA cleavage, but not for ssDNA cleavage. This also explains why mismatches within the seed region abolish dsDNA cleavage, and yet are tolerated for ssDNA cleavage by Cas12i2, indicating that the PAM and complementarity within the seed region are not required for ssDNA cleavage. However, the pairing in the central spacer region (nucleotides 6–15) is required for the activation of RuvC catalytic pocket, because it interacts directly with and affects the conformation of the Helical-II domain. The sequence and structural similarities between Cas12i and Cas12b indicate that complementary target ssDNA binding likely also activates Cas12b and triggers its non-specific single-stranded DNase activity. Similarly, complementarity between the crRNA and target DNA within the seed region is likely not crucial for *trans* ssDNA cleavage by both Cas12a and Cas12b.

Metal ions have essential roles for DNA cleavage by the RuvC domain in Cas12 and Cas9. In a previous study of AacCas12b, a short DNA substrate was observed in the RuvC active site of a catalytically inactive variant (Fig. 7a)[9]. However, no metal ions were observed in these AacCas12b structures. Our Cas12i2$^{wt}$–crRNA–DNA ternary complex with two metal ions bound in the RuvC catalytic pocket represents a substrate binding state of the catalytic pocket (Fig. 7b).

The Cas12i2 RuvC domain shares a similar fold and catalytic site with RNase H and integrase proteins[31,32]. Their active pockets are primarily composed of negatively charged carboxylate residues, which coordinate two divalent metal ions. In the two metal mechanism common to these enzymes, $Mg^{2+}_A$ assists the

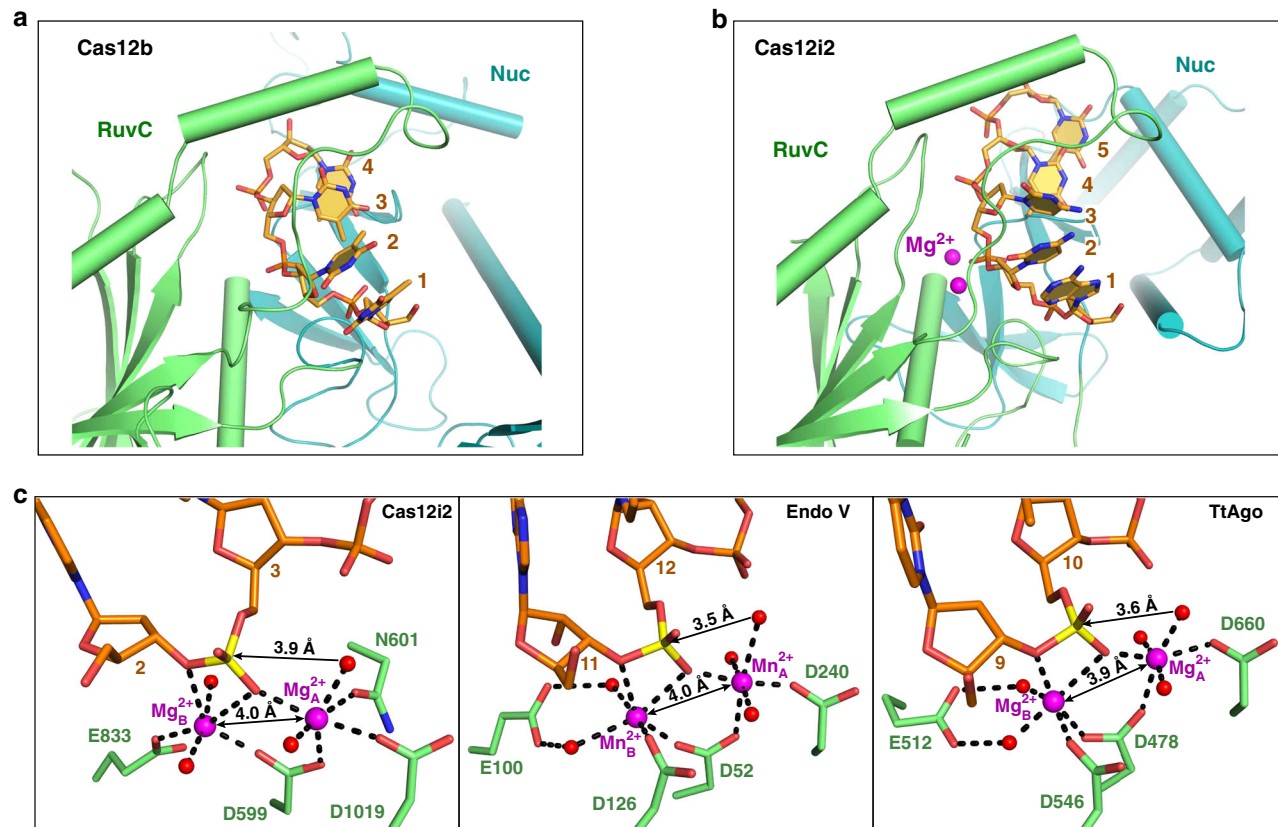

**Fig. 7 Structural comparison of Cas12i2, Cas12b, and other RNase H fold-containing proteins. a, b** Comparison of the ssDNA substrate and the RuvC domain in AacCas12b (**a**) and Cas12i2 (**b**). **c** The substrate and two metal cations in the catalytic pocket of Cas12i2 (left panel), mouse Endonuclease V (center panel, PDB ID: 6OZL), and TtAgo (right panel, PDB ID: 4NCB).

nucleophilic attack by positioning and activating a water molecule, while $Mg^{2+}_B$ stabilizes the transition state and leaving group[33]. Two metal ions separated by a distance of 3.9–4.0 Å were also observed in RNase H1, mouse Endonuclease V, and TtAgo proteins (Fig. 7c)[27,32,34], suggesting that two metal ions separated by ~4 Å is a common feature of the substrate binding state of RuvC and RNase-containing enzymes. Upon the water nucleophile attacking the scissile phosphate, these two metal ions move closer, and as a result, generate a pentacovalent intermediate[32].

In conclusion, our structures of the Cas12i2 binary and ternary complexes provide insight into the molecular mechanisms of crRNA-mediated targeting and cleavage of DNA by the type V-I CRISPR-Cas system. We show the structure of RuvC in the active state, revealing how two metals catalyze DNA cleavage by all Cas9 and Cas12 proteins. Our biochemical studies show that Cas12i2 cleaves dsDNA in a PAM sequence-dependent manner, and ssDNA in a PAM-independent manner. Furthermore, the formation of the crRNA:target DNA hybrid activates the RuvC catalytic pocket, with base-pairing in the seed region not being essential for activation. Our findings have implications for developing more reliable genome-editing or nucleic acids detection tools.

## Methods

**Cas12i2 protein expression and purification.** Full-length ORF of Cas12i2 (encoding residues 1–1054) was synthesized from Sangon Biotech and was cloned into the expression vector pET28a-Sumo containing a His$_6$-Sumo tag at the N terminus. Mutants were constructed using a site-directed mutagenesis kit. All proteins were overexpressed in *E. coli* BL21 (DE3) (Novagen) cells and were induced with 0.1 mM isopropyl-β-D-1-thiogalactopyranoside (IPTG) at OD$_{600}$ = 0.6 for 12 h at 18 °C.

Cells expressing Cas12i2 were harvested and then disrupted by sonication in buffer containing 20 mM Tris-HCl (pH 7.5) and 0.5 M NaCl at 4 °C following high speed centrifugation. After centrifugation, the supernatant was incubated with $Ni^{2+}$-Sepharose resin (GE Healthcare), and the bound protein was eluted with lysis buffer supplemented with 200 mM imidazole. His$_6$-Sumo tag was cleaved from Cas12i2 protein by His-tagged ubiquitin-like protein 1 (Ulp1) protease during dialysis against buffer containing 20 mM Tris-HCl (pH 7.5), 0.4 M NaCl for 2 h at 4 °C. Cas12i2 protein was further purified by $Ni^{2+}$-Sepharose column to remove the His$_6$-Sumo tag. The flow-through collections were loaded into a Heparin column (GE Healthcare), and Cas12i2 was eluted with buffer containing 20 mM Tris-HCl (pH 7.5) and 1 M NaCl. The seleno-methionine (SeMet) substituted Cas12i2 was expressed and purified using the same protocol.

**In vitro transcription and purification of crRNA and pre-crRNA.** The crRNA and 61-nt pre-crRNA were synthesized by in vitro transcription using a T7 RNA polymerase and linearized plasmid DNA templates. Transcription reactions were performed at 37 °C for 4 h with 100 mM HEPES-KOH (pH 7.9), 25 mM MgCl$_2$, 30 mM DTT, 2 mM each NTP, 2 mM spermidine, 0.1 mg/ml T7 RNA polymerase, and 40 ng/μl linearized plasmid DNA template. The RNAs were purified by gel electrophoresis on a 20% denaturing (8 M urea) polyacrylamide gel and an Elutrap system. Finally, RNA was resuspended in DEPC (diethylpyrocarbonate) H$_2$O and stored at −80 °C. All the RNA sequences are listed in the Supplementary Table 1.

**Preparation of DNA.** All short DNAs used in this study were purchased from Sangon Biotech. All DNAs were dissolved in buffer containing 20 mM Tris-HCl (pH 7.5), 150 mM NaCl, 10 mM MgCl$_2$. For double-stranded DNA, equimolar amounts of two complementary strands were pre-denatured at 95 °C for 10 min and then annealed at room temperature. All the DNA sequences are listed in the Supplementary Table 1.

**Reconstitution of Cas12i2–crRNA binary and Cas12i2–crRNA-dsDNA ternary complexes.** The Cas12i2–crRNA complex was reconstituted by mixing Cas12i2 with crRNA at a ratio of 1:1.05 on ice for 30 min in buffer containing 20 mM Tris-HCl (pH 7.5), 300 mM NaCl. The Cas12i2–crRNA–dsDNA complexes were reconstituted by incubating Cas12i2 with crRNA first on ice for 30 min, and

dsDNA was added for incubation for another 30 min. The molar ratios of Cas12i2, crRNA, and dsDNA were 1:1.1:1.3. The complexes were purified by gel filtration with Superdex 200 increase 10/300 column (GE Healthcare) in buffer containing 20 mM Tris-HCl (pH 7.5), 300 mM NaCl. Finally, the binary and ternary Cas12i2 complexes were concentrated to $A_{280}$ absorbance of 5.5 and 7.8, respectively, as measured by Nanodrop 2000, before crystallization.

**Crystallization, data collection, and structure determination.** All crystals were grown by mixing 1 μl Cas12i2 complex with 1 μl reservoir solution by hanging-drop vapor-diffusion method and incubated at 16 °C. Crystals of the Cas12i2–crRNA binary complex were grown from 0.14 M sodium citrate, 73 mM Tris-HCl (pH 8.2), 27 mM HEPES (pH 7.5), 13% PEG 3350, 4.5% PEG 20000. Crystals of the Cas12i2$^{wt}$–crRNA-26 nt dsDNA ternary complex were grown from buffer containing 0.2 M NaCl, 0.1 M Tris-HCl (pH 7.2), 18% PEG 6000. Crystals of the Cas12i2$^{E833A}$–crRNA-26 nt dsDNA complexes were grown from 0.2 M ammonium acetate, 20% PEG 3350. Crystals of Cas12i2$^{E833A}$–crRNA-12-nt dsDNA were grown from 0.2 M NaNO$_3$, 0.1 M Bis-Tris propane (pH 7.8), 17% PEG 6000. All crystals above were cryoprotected using the corresponding reservoir solution supplemented with ethylene glycol, and flash-frozen in liquid nitrogen.

All diffraction datasets were collected at beamline BL17U1 or BL19U1 at the Shanghai Synchrotron Radiation Facility (SSRF), or at BL41XU of Spring-8, and processed with XDS or HKL2000[35]. The phase of a structure of Cas12i2$^{E833A}$–crRNA–dsDNA was solved with the Se single wavelength anomalous dispersion (SAD) method using PHENIX Autosol[36,37]. After the initial phases were obtained, the atomic models were manually built and adjusted using the program COOT[38]. Other structures of Cas12i2 were solved by molecular replacement (MR) using PHENIX PHASER[39], and the Cas12i2$^{E833A}$–crRNA–dsDNA complex structure was used as the starting model. Iterative cycles of crystallographic refinement were performed using PHENIX[36,37]. All structural figures were prepared using the PyMOL. Data collection and refinement statistics are listed in Table 1. Ramachandran statistics were generated using MolProbity[40].

**Target cleavage assays.** The in vitro target dsDNA cleavage assays were performed using linearized pUC19 plasmid targets. Cas12i2 was incubated with crRNA at a molar ratio of 1:1.1 first for 30 min on ice. 200 nM Cas12i2–crRNA was incubated with 300 ng pUC19 target DNA at 37 °C for 15 min in 10 μl reactions containing 20 mM Tris-HCl (pH 7.5), 100 mM KCl, 10 mM MgCl$_2$, 1 mM DTT and 5% glycerol. For metal ion assays, MgCl$_2$ was omitted from the buffer and substituted with EDTA or different metal ions such as Mn$^{2+}$, Ca$^{2+}$, Ni$^{2+}$, Co$^{2+}$, Zn$^{2+}$, Cd$^{2+}$, and Cu$^{2+}$ at a concentration of 5 mM. For the cleavage of linear plasmid by pre-formed Cas12i2–crRNA-target ssDNA, 200 nM Cas12i2–crRNA at a molar ratio of 1:1 was prepared on ice for 30 min first, and then 200 nM target ssDNA without PAM and 300 ng linear plasmid were added to the reactions simultaneously. Reactions were incubated at 37 °C for 15 or 45 min. All reactions above were stopped by adding 2 μl gel-loading dye containing 10 mM EDTA (NEB). The reaction products were analyzed on 1% agarose gels stained with ethidium bromide for products visualization. All experiments were carried out at least in triplicate, with representative replicates shown in the figure panels.

**Characterization of specific cleavage sites.** In a 50 μl reaction system, 300 nM Cas12i2 was incubated with 330 nM crRNA for 30 min on ice, and then 2.7 μg linear pUC19 target plasmids were introduced. The reaction was carried out at 37 °C for 30 min in buffer containing 20 mM Tris-HCl (pH 7.5), 100 mM KCl, 10 mM MgCl$_2$, 1 mM DTT and 5% glycerol and stopped by adding gel-loading dye containing 10 mM EDTA (NEB). Cleavage products were separated by 1% Agarose gel, and the two product bands were extracted by gel extraction kit (Omega). Subsequently, the two products were verified by Sanger sequencing. All experiments were carried out at least in triplicate, with representative replicates shown in the figure panels.

**ssDNA trans-cleavage assays.** A 5′-Cy3-labeled 53-nt ssDNA was employed as the substrate for the ssDNA trans cleavage by Cas12i2. To form the Cas12i2–crRNA binary complex, Cas12i2 and crRNA were mixed on ice at a 1:1.1 molar ratio. Next, 200 nM pre-formed Cas12i2–crRNA complex, 200 nM Cy3-labeled ssDNA substrate, and reaction buffer (20 mM Tris-HCl (pH 7.5), 100 mM KCl, 10 mM MgCl$_2$, 1 mM DTT and 5% glycerol) were mixed in 10 μl reactions systems, followed by the additional of 200 nM DNA activator. All reactions above were carried out at 37 °C for 15 min. Reactions were stopped by adding 10 μl 2× urea loading buffer and 95 °C quenched for 10 min. Products were separated by 20% urea denaturing polyacrylamide gels and visualized using a FluorChem system (ProteinSimple).

For comparison of activation of trans-cleavage activity by target ssDNA and ssRNA, a binary enzyme complex of Cas12i2–crRNA at a molar ratio of 1: 1.1 were prepared on ice for 15 min first. In 10 μl reaction system, a gradient of enzyme concentrations was set up: 100, 300, 500, 800, 1000 and 2000 nM. Thereafter, 200 nM Cy3-labeled 53-nt non-target substrate was added, Finally, target ssDNA or ssRNA without PAM was added to the enzyme at a molar ratio of 1: 1 and reactions were carried out at 37 °C for 15 min. All experiments were carried out at least in triplicate, with representative replicates shown in the figure panels.

**Pre-crRNA processing assays.** For pre-crRNA processing experiments, 20 μM Cas12i2 was incubated with 20 μM pre-crRNA substrate, and divalent cations as indicated at 37 °C for 30 min. Processing reactions (total volume of 10 μl) contained 20 mM Tris-HCl (pH 7.5), 100 mM KCl, 5% glycerol, and 5 mM chelating agent (EDTA) or divalent cation (MgCl$_2$, MnCl$_2$, CaCl$_2$).

Reactions were stopped by adding 1 μl 1 mg/ml Proteinase K and incubated at 37 °C for 30 min. Subsequently, 10 μl 2× urea loading buffer was added for incubation at 65 °C for 10 min. For all the reactions, 20% urea denaturing polyacrylamide gel with TBE buffer was used for analysis. Cleavage products were visualized by toluidine blue staining. All experiments were carried out at least in triplicate, with representative replicates shown in the figure panels.

**Reporting summary.** Further information on research design is available in the Nature Research Reporting Summary linked to this article.

## Data availability
The crystallographic datasets reported herein have been deposited in the PDB repository under accession codes 6LTU Cas12i2$^{WT}$-crRNA-dsDNA (26 nt), 6LTR Cas12i2$^{E833A}$-crRNA-dsDNA (26 nt), 6LU0 Cas12i2$^{E833A}$-crRNA-dsDNA (12 nt), 6LTP (Cas12i2$^{WT}$-crRNA). The data that support the findings of this study are available from the corresponding authors on reasonable request. Source data are provided with this paper.

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

## Acknowledgements

We are grateful to the staff of the BL-17U1 and BL-19U1 beamlines at the Shanghai Synchrotron Radiation Facility, and the BL41XU beamline at SPring-8 (2019A2533). This work was supported by grants from the Chinese Ministry of Science and Technology (2017YFA0504203), the Natural Science Foundation of China (91940302, 31725008, 31630015, 31930065, and 31700662), and the Chinese Academy of Sciences (XDB37010202 and QYZDY-SSW-SMC021). We thank Dr. Guy Riddihough in Life Science Editors for help with editing the manuscript, and Xudong Zhao and Xiaofei Guo in Institute of Biophysics for offering the facility equipment support.

## Author contributions

X.H. expressed and purified the proteins and grew all crystals. M.C., X.L., and J.W. collected X-ray diffraction data, and Z.C. solved all of the structures. W.S. and X.H. performed the biochemical assays. Y.W. wrote the manuscript. G.S., W.G., and Y.W. edited the manuscript. Y.W. supervised all of the structural and biochemical studies.

## Competing interests

The authors declare no competing interests.
