## [Peer Review File · Nature Communications]

REVIEWER COMMENTS

Reviewer #1 (Remarks to the Author):

Huang et al. present several structures of Cas12i2: in complex with guide RNA (crRNA) and in complex with crRNA and various dsDNA targets. The obtained structures in combination with activity assays and mutational analysis allow authors to propose molecular mechanisms of crRNA processing, of DNA target recognition and of DNA cleavage through activation of the RuvC domain.

Overall, this is an important study that not only explains the molecular mechanism of Cas12i2 mediated DNA cleavage but also provides significant insights into the DNA cleavage mechanism by RuvC-containing Cas proteins. The data is technically sound and provides strong evidence for the conclusions presented in this study. This work is of interest to many, because the proposed molecular mechanisms have implications for developing more reliable genome-editing and nucleic acids detection tools. Although I am very enthusiastic about this work, I have some concerns that I hope the authors will be able to address in a revised manuscript.

Major concerns:

1. My main concern is the structure of the results section of the manuscript. Currently it reads as a set of disconnected paragraphs. The authors don't provide a smooth story, which diminishes the impact of the work performed here. I would suggest first presenting the structure of Cas12i2 in complex with crRNA and discussing:

- a. the basis of crRNA maturation (where is the catalytic center, what are the resulting crRNA termini);
- b. how is the 5' handle of crRNA recognized by Cas12i2;
- c. how is the spacer recognized;
- d. how the length of the spacer affects Cas12i2 activity (what is the optimal length for dsDNA cleavage, does the length of the spacer affect the kinetics and order of target and non-target strand cleavage);
- e. explain how the Cas12i2-crRNA complex is competent for target search but not for trans ssDNA cleavage.

Then I would suggest presenting the Cas12i2-crRNA-DNA complexes:

- a. the structure overall and what the differences are between secondary and tertiary complexes;
- b. PAM recognition;
- c. crRNA-TS DNA hybrid recognition;
- d. NTS conformation;
- e. Cas12i2 activation for trans cleavage.

2. When discussing structures, it would be helpful if authors reference PDB IDs in the text and in the figure captions. As is now, it is confusing which structures are referenced exactly.

3. Extended Data fig. 1g: From the figure it is not clear which residues have effect on dsDNA cleavage. Quantification is also needed. Moreover, authors should include the details of the cleavage reaction in the figure caption (concentrations of components, reaction time etc.).

4. Page 7-8: "...the bases of the PAM duplex form extensive indirect 8 hydrogen bonds via water molecules with either Cas12i2 or other bases within the PAM duplex (Fig. 2h)." Which Cas12i2 residues are important for PAM recognition? Does the interaction of Cas12i2 with the PAM affect the conformation of the target strand? It is not clear how Cas12i2 interaction with PAM promotes dsDNA unwinding.

5. Page 12: "Thus, RuvC catalytic pocket is inactive prior to the formation of the guide:target heteroduplex, which induces a rearrangement of the Helical-II domain and activates the RuvC catalytic pocket." Did authors try to delete Helical-II domain to obtain constantly active Cas12i2?

6. Page 13: "In contrast, non-complementary ssDNA and fully duplexed dsDNA failed to activate the ssDNA cleavage of Cas12i2." This statement is confusing. It's not clear if dsDNA has complementarity to the crRNA.

7. Authors should include a figure of the structure depicting the catalytic center of crRNA processing. Is it in the RuvC domain? Is it the same as the active center of DNA cleavage? Does the inactivation of crRNA processing center affect DNA cleavage?

8. It would be interesting to see if crRNA processing has effect on DNA cleavage kinetics. E.g. if the Cas12i2-crRNA and Cas12i2-pre-crRNA complexes exhibit different DNA cleavage kinetics.

9. More systematic structural and biochemical comparison of Cas12i2 with the Cas12a, Cas12b, Cas12j (CasPhi), Cas12e (CasX), Cas9 needed to clarify and emphasize the differences and peculiarities of Cas12i2.

10. Authors should include the details of activity assays (concentrations of components, reaction time points etc.) in the figure captions of Fig. 2 (d/e/i); Fig. 3 (c/e); Fig. 5 (a/b/c/d/e); Extended Data fig. 1g; Extended Data fig. 3; and Extended Data fig. 6.

Minor concerns:

1. Page 5: "The RuvC and Nuc domains are positioned side-by-side, forming a positively charged concavity, in which an ssDNA is bound." It would be helpful if authors added a supplementary figure demonstrating the charge distribution on the structures.

2. Page 5: "One side of the WED domain contacts the 23-nt crRNA repeat and the opposite site interacts with the dsDNA." Could author reference the figures demonstrating WED domain interactions?

3. Page 6: "This heteroduplex lies in the central channel formed by REC and NUC lobes and is recognized by Cas12i2 in a sequence-independent manner (Extended Data Fig. 2)." This sentence is confusing. I do not know any Cas proteins that recognize the sequence of the RNA-DNA duplex. The complementarity between crRNA and DNA is usually recognized instead.

4. Page 15: "...which cleaves dsDNA sequence specifically and ssDNA non-specifically." This sentence is not clear. There are no known Cas effectors that cleave dsDNA in a sequence dependent manner. The target is selected through the complementarity to the crRNA.

Reviewer #2 (Remarks to the Author):

Dear editor, dear authors,

CRISPR-Cas effector enzymes use CRISPR RNA guides to bind and cleave complementary DNA targets. Type V CRISPR-Cas effector enzymes, including Cas12a-Cas12i, either are used for programmable genome engineering (especially Cas12a), or demonstrate high potential to expand to genome editing toolbox due to smaller size, different PAM requirements, and/or different molecular mechanisms. While all Cas12 effectors belong to Type V CRISPR-Cas systems, they have followed different paths of evolution and therefore have not only low sequence similarities, but also distinct domain compositions.

While detailed insights into the structures and mechanism of especially Cas12a and Cas12b have been obtained during the last years, little is known about the more recently discovered Cas12 variants. Furthermore, experimental evidence for the exact catalytic mechanism by which the RuvC domain – which appears conserved throughout all Cas12a variants – can cleave the target DNA has remained elusive.

In the manuscript entitled "Structural basis for two metal-ion catalysis of DNA Cleavage by Cas12i2", Huang et al. have performed a detailed study in which they have solved a total of six structures of the Cas12i variant Cas12i2. These structures do not only reveal the domain composition and architecture of this Type V effector enzyme, but additionally give important insights into how Cas12i2 recognizes its crRNA guide, how it recognizes the PAM in dsDNA targets, how dsDNA targets are unwound prior to crRNA-DNA target strand hybridization, and how crRNA-target DNA binding induces conformational changes that eventually result in RuvC activation. Of note, Huang et al. demonstrate – for the first time (to my knowledge) – the catalytic mechanism by which Type V RuvC domains cleave DNA molecules. The structural data is extended with various biochemical studies that corroborate the structural observations. This paper generates unprecedented insights into the structure and mechanism of Cas12i2, and into the Type V RuvC domain and activity in general.

I highly recommend publication after the authors have addressed some (minor) comments. Most of my comments can (most likely) be addressed textually or by further structure refinement.

-The reported completeness in the PDB validation reports is much lower (around 80-85%) for 6LTP, 6LTR, and 6LTX, while it is close to 100% in Table 1. How can this discrepancy be explained?

-There is a substantial amount of Ramachandran outliers (and a relative high amount of Ramachandran allowed) in each of the structures. Similarly, there is a high amount of rotamer outliers for 6LTP, 6LTR, 6LTU, 6LTX, and 6LU0. Can these Ramachandran and rotamer outliers be explained, or should the structures be refined further?

-Throughout the manuscript, the authors use the terms spacer and repeat for the crRNA. The spacer and repeat are located in the CRISPR locus. Instead, the crRNA contains a spacer-derived segment and a repeat-derived segment. To prevent confusion, the authors should either change this throughout the manuscript (preferred), or indicate once in the beginning of the manuscript (and preferentially in Figure 1) what they mean with the spacer and repeat in the crRNA.

-It is not clear to me what the source (and/or sequence) of the DNA in the catalytic site is. Can the authors explain this, and explain based on what they have based the sequence in the PDB files? Is this sequence an assumption or can it be deduced with certainty from the density?.

-How do the authors explain that in the structure intact DNA is observed in the catalytic site, while the Cas12i2 is in the cleavage-competent state?

-Can the authors explain if the Helical-II domain and/or the loop that is inserted into the RuvC catalytic pocket is conserved throughout Type V effectors? Has this been described in other papers? For example, can this loop be compared to the 'lid' described for Cas12a (Stella et al. 2018)? If it is conserved, this would not only reveal the activation of Cas12i2, but potentially of other Type V effectors too.

Minor (textual) comments

Below, I have listed various comments that I think will improve readability and will make it easier to understand the figures.

-In my opinion, both the abstract and introduction could do more justice to both earlier findings about Cas12i2 activity (Yan et al.) and to the importance of the work described in this manuscript. The abstract jumps to the knowledge gap and obtained results directly. An introduction would help to put these important findings into context. Furthermore, it is unclear how the seed region and central spacer are defined in the abstract.

In the introduction, it should be clarified here that Cas12i is most closely related to Cas12b based on phylogeny, but resembles Cas12a in the fact that it also relies on a single guide RNA (and not a tracrRNA). Furthermore, more is known about its cleavage activity (i.e. it has been shown that it does

not efficiently cleave ssDNA targets in cis and demonstrates collateral ssDNA cleavage in trans (Yan et al.)) and that should be noted.

Page 3:

-Class 2 CRISPR-Cas systems also consist of Cas1, Cas2 and crRNA. It should be rephrased stating that Class 2 rely on a single large protein-crRNA complex for CRISPR interference.

-The authors should explain why RuvC cleaves ssDNA and not dsDNA, as Cas9 and Cas12a both can cleave dsDNA targets.

Page 4:

-it should be made clear that Cas12a is expected to cleave the DNA similar to Cas12b (unless the authors disagree based on their findings), as both rely on a single RuvC catalytic site for target cleavage.

Page 5:

-how identical are the two structures? RMSD is 0.0?

-What is the sequence identity between Cas12i2 and AacCas12b? Are there domains in these two enzymes which are more similar and domains which are less similar?

-Please indicate exactly how many nucleotides of the crRNA and DNA are visible (e.g. 45 out of 54). In fact, it would be insightful to include a supplemental figure displaying all solved structures and the sequences of the RNA and DNA molecules that they have bound, and color-code whether certain nucleotides in these sequences are visible in the structure or not.

Page 8:

-RuvC is not the sole catalytic domain in Cas12i2 – it might be the sole DNA-cleaving catalytic domain, but the authors clearly demonstrate there is a RNA-cleaving catalytic domain too.

Figure 5a:

-The PAM colouring scheme is counterintuitive to me. Can the authors change the colouring scheme or indicate what colours indicates absence and presence of the PAM/mismatches.

With kind regards,
Daan Swarts

Response to reviewers:

We thank the reviewers for their time and helpful comments on the manuscript. We have made revisions (**highlighted in red**) to the manuscript to address their concerns. The responses to the reviewers' comments are below (**in blue**).

REVIEWER COMMENTS

Reviewer #1 (Remarks to the Author):

Huang et al. present several structures of Cas12i2: in complex with guide RNA (crRNA) and in complex with crRNA and various dsDNA targets. The obtained structures in combination with activity assays and mutational analysis allow authors to propose molecular mechanisms of crRNA processing, of DNA target recognition and of DNA cleavage through activation of the RuvC domain.

Overall, this is an important study that not only explains the molecular mechanism of Cas12i2 mediated DNA cleavage but also provides significant insights into the DNA cleavage mechanism by RuvC-containing Cas proteins. The data is technically sound and provides strong evidence for the conclusions presented in this study. This work is of interest to many, because the proposed molecular mechanisms have implications for developing more reliable genome-editing and nucleic acids detection tools. Although I am very enthusiastic about this work, I have some concerns that I hope the authors will be able to address in a revised manuscript.

We thank the reviewer for their positive comments and for summarizing the important findings in our work.

Major concerns:

1. My main concern is the structure of the results section of the manuscript. Currently it reads as a set of disconnected paragraphs. The authors don't provide a smooth story, which diminishes the impact of the work performed here. I would suggest first presenting the structure of Cas12i2 in complex with crRNA and discussing:

a. the basis of crRNA maturation (where is the catalytic center, what are the resulting crRNA termini);

b. how is the 5' handle of crRNA recognized by Cas12i2;

c. how is the spacer recognized;

d. how the length of the spacer affects Cas12i2 activity (what is the optimal length for dsDNA cleavage, does the length of the spacer affect the kinetics and order of target and non-target strand cleavage);

e. explain how the Cas12i2-crRNA complex is competent for target search but not for trans ssDNA cleavage.

Then I would suggest presenting the Cas12i2-crRNA-DNA complexes:

- a. the structure overall and what the differences are between secondary and tertiary complexes;*
- b. PAM recognition;*
- c. crRNA-TS DNA hybrid recognition;*
- d. NTS conformation;*
- e. Cas12i2 activation for trans cleavage.*

We thank the reviewer for the suggestion, but we prefer to keep the original text organization in the revision owing to the following reasons. First, the structures of RuvC-containing Cas complexes in their catalytically competent states, with both metal ions and the ssDNA, are crucial to understand how the RuvC catalytic domain of Class 2 Cas proteins and this is the first Cas12 structure with RuvC in the catalytic state. This is the key point of our manuscript, and we prefer to present the most important result first for the clarity. Second, the resolution of the Cas12i2-crRNA-dsDNA ternary complex (2.57 Å) is much higher than that of the Cas12i2-crRNA binary complex (3.4 Å). The higher resolution structures are able to provide better data to analyze the interactions between crRNA and Cas12i2. In addition, our biochemical studies and structure of Cas12i2-crRNA-dsDNA ternary complex are able to clearly demonstrate the crRNA recognition, the effect of the space on the Cas12i2 activity and target search.

2. When discussing structures, it would be helpful if authors reference PDB IDs in the text and in the figure captions. As is now, it is confusing which structures are referenced exactly.

We thank the reviewer for the suggestion. PDB IDs have been included in the text and figure captions in the revised version.

3. Extended Data fig. 1g: From the figure it is not clear which residues have effect on dsDNA cleavage. Quantification is also needed. Moreover, authors should include the details of the cleavage reaction in the figure caption (concentrations of components, reaction time etc.).

We thank the reviewer for this suggestion.

The details of the cleavage reaction have been included in some figure captions in the revised version. Due to space constraints, the specific conditions of the reaction are given only when the same type of reaction appears for the first time.

In contrast to the previously determined structures of crRNA-bound Cas proteins (Cas9, Cas12a, Cas12b, and Cas13a) that form a few hydrogen bonds with the bases of the repeat region, Cas12i2 forms much more base contacts with the crRNA repeat. Numerous residues of Cas12i2 form hydrogen bonds with bases of the repeat, resulting in that a single point mutation exhibits slight effect on DNA cleavage activity. Therefore, except for the mutants

K862A and K845A, most of the mutants in Fig. 1g (Extend Data Fig 2c in the current version) do not display detectable effect on the DNA cleavage.

4. Page 7-8: “...the bases of the PAM duplex form extensive indirect 8 hydrogen bonds via water molecules with either Cas12i2 or other bases within the PAM duplex (Fig. 2h).” Which Cas12i2 residues are important for PAM recognition? Does the interaction of Cas12i2 with the PAM affect the conformation of the target strand? It is not clear how Cas12i2 interaction with PAM promotes dsDNA unwinding.

To address the reviewer’s concerns about the residues important for PAM recognition, we conducted experiments using a cleavage assay with the mutants of residues that form hydrogen bond with the PAM sequence via water. We found that the DNA cleavage was severely impaired or completely abrogated by mutation of residues that interact with the PAM sequence. These results are now incorporated into the Fig. 3d of the revised manuscript.

The hydrogen bonds between the bases of PAM and Cas12i2 likely destabilize the Watson-Crick base pairing adjacent to the PAM. These interactions may result in changes in the DNA structure and promoting the unwinding of dsDNA upon Cas12i2 recognize the PAM sequence, suggesting that the interactions between the PAM and Cas12i2 are essential for dsDNA unwinding. Similarly, Cas9-sgRNA complex binds to the PAM sequence favors separation of a few PAM-proximal protospacer base pairs (Mekler, PNAS, 2017). Furthermore, residues N164 and Q163 of Cas12i2 stack on the base-pair (-1) of dsDNA, preventing the further base-pairing of the target and non-target strands beyond position (-1). These interactions facilitate the hydride of the target strand and the guide and forming the R-loop conformation.

5. Page 12: “Thus, RuvC catalytic pocket is inactive prior to the formation of the guide:target heteroduplex, which induces a rearrangement of the Helical-II domain and activates the RuvC catalytic pocket.” Did authors try to delete Helical-II domain to obtain constantly active Cas12i2?

The guide:target hydride induces the conformational change of the Helical-II domain, which rotates away from the RuvC domain upon the formation of the guide:target heteroduplex, generating a wider binding channel for the guide:target duplex. The rotation of the Helical-II domain results in the RuvC catalytic pocket accessible for the ssDNA substrate, activating the RuvC catalytic pokcet.

The Helical-II domain that forms multiple interactions with the guide:target duplex, is crucial for the DNase activity of the Cas12i2. The deletion of the Helical-II domain likely destabilize the guide:target duplex, resulting in a inactive Cas12i2. Thus, we didn't make a Cas12i2 mutant without the Helical-II domain.

6. Page 13: *“In contrast, non-complementary ssDNA and fully duplexed dsDNA failed to activate the ssDNA cleavage of Cas12i2.” This statement is confusing. It’s not clear if dsDNA has complementarity to the crRNA.*

We thank the review for pointing out this issue.

One strand of the dsDNA is complementary to the crRNA. We rewrote this sentence in the revision.

7. *Authors should include a figure of the structure depicting the catalytic center of crRNA processing. Is it in the RuvC domain? Is it the same as the active center of DNA cleavage? Does the inactivation of crRNA processing center affect DNA cleavage?*

We thank this reviewer for this suggestion.

The crRNA processing catalytic site is located in the Helical-I and WED domains, which differs from the RuvC DNase catalytic pocket. The figure depicting the catalytic center of the crRNA processing is included in the revised manuscript as suggested (Extended Data Fig. 6e).

To test whether the inactivation of crRNA processing center affect DNA cleavage by the RuvC catalytic pocket, we performed the dsDNA cleavage assay using the pre-crRNA and the mutants of residues crucial for crRNA processing. We found that the mutation of amino acids Lys18, His485 or His486 decreases the DNA cleavage by Cas12i2-pre-crRNA complex (Extended Data Fig. 6f).

8. *It would be interesting to see if crRNA processing has effect on DNA cleavage kinetics. E.g. if the Cas12i2-crRNA and Cas12i2-pre-crRNA complexes exhibit different DNA cleavage kinetics.*

We thank the reviewer for this suggestion.

To test the effect of the crRNA processing on DNA cleavage kinetics, we compared the DNA cleavage activity by the wild type Cas12i2 or the mutants of His485, His486 or Lys18 by incubating with either mature crRNA or pre-crRNA. We found that H485A, H486A and K18A mutants significantly decrease the Cas12i2-pre-crRNA mediate DNA cleavage. By contrast, these mutants display no detectable effect on the Cas12i2-crRNA complex mediated DNA cleavage (Extended Data Fig. 6f).

9. *More systematic structural and biochemical comparison of Cas12i2 with the Cas12a, Cas12b, Cas12j (CasPhi), Cas12e (CasX), Cas9 needed to clarify and emphasize the differences and peculiarities of Cas12i2.*

We thank this reviewer for this suggestion. We compared the structure and biochemistry of Cas12i2 with the Cas12a, Cas12b in the discussion with a new figure (Extended Data Fig. 7). We also like to systematically compare the structure and biochemistry of Cas12i2 with the Cas12j, Cas12e (CasX), and Cas9 in our manuscript. However, the main text of the article in *Nature Communications* is limited to 5,000 words, and our revision has no more space for further comparison of distinct Cas12 effectors.

10. Authors should include the details of activity assays (concentrations of components, reaction time points etc.) in the figure captions of Fig. 2 (d/e/i); Fig. 3 (c/e); Fig. 5 (a/b/c/d/e); Extended Data fig. 1g; Extended Data fig. 3; and Extended Data fig. 6.

We thank the reviewer for the suggestion.

The details of activity assays are included in some revised figure captions. Due to space constraints, the specific conditions of the reaction are given only when the same type of reaction appears for the first time. However, all details of the biochemical assays are available in the Method details.

Minor concerns:

1. Page 5: "The RuvC and Nuc domains are positioned side-by-side, forming a positively charged concavity, in which an ssDNA is bound." It would be helpful if authors added a supplementary figure demonstrating the charge distribution on the structures.

Supplementary figure (Extended Data Fig. 4a) was added as suggested.

2. Page 5: "One side of the WED domain contacts the 23-nt crRNA repeat and the opposite site interacts with the dsDNA." Could author reference the figures demonstrating WED domain interactions?

In the revised manuscript, Fig. 1c was included after the sentence "One side of the WED domain contacts the 23-nt crRNA repeat and the opposite site interacts with the dsDNA".

3. Page 6: "This heteroduplex lies in the central channel formed by REC and NUC lobes and is recognized by Cas12i2 in a sequence-independent manner (Extended Data Fig. 2)." This sentence is confusing. I do not know any Cas proteins that recognize the sequence of the RNA-DNA duplex. The complementarity between crRNA and DNA is usually recognized instead.

We thank the review for pointing out this issue.

We replaced this sentence by the sentence "This heteroduplex lies in the central channel formed by REC and NUC lobes and is stabilized by the hydrogen-bonding and ionic interactions between backbone sugar-phosphate groups and the Helical-I-III, RuvC and WED domains of Cas12i2 (Extended Data Fig. 3). "

4. Page 15: “...which cleaves dsDNA sequence specifically and ssDNA non-specifically.”
This sentence is not clear. There are no known Cas effectors that cleave dsDNA in a sequence dependent manner. The target is selected through the complementarity to the crRNA.

The sentence has been changed as suggested. In the revised manuscript, we replaced the original sentence by “...which cleaves the dsDNA complementary to crRNA in cis and ssDNA in trans”.

Reviewer #2 (Remarks to the Author):

While detailed insights into the structures and mechanism of especially Cas12a and Cas12b have been obtained during the last years, little is known about the more recently discovered Cas12 variants. Furthermore, experimental evidence for the exact catalytic mechanism by which the RuvC domain – which appears conserved throughout all Cas12a variants – can cleave the target DNA has remained elusive.

In the manuscript entitled “Structural basis for two metal-ion catalysis of DNA Cleavage by Cas12i2”, Huang et al. have performed a detailed study in which they have solved a total of six structures of the Cas12i variant Cas12i2. These structures do not only reveal the domain composition and architecture of this Type V effector enzyme, but additionally give important insights into how Cas12i2 recognizes its crRNA guide, how it recognizes the PAM in dsDNA targets, how dsDNA targets are unwound prior to crRNA-DNA target strand hybridization, and how crRNA-target DNA binding induces conformational changes that eventually result in RuvC activation. Of note, Huang et al. demonstrate – for the first time (to my knowledge) – the catalytic mechanism by which Type V RuvC domains cleave DNA molecules. The structural data is extended with various biochemical studies that corroborate the structural observations. This paper generates unprecedented insights into the structure and mechanism of Cas12i2, and into the Type V RuvC domain and activity in general.

I highly recommend publication after the authors have addressed some (minor) comments. Most of my comments can (most likely) be addressed textually or by further structure refinement.

We are grateful to Dr. Swarts for his positive feedback, especially for highlighting that our manuscript generates unprecedented insights into the structure and mechanism of Cas12i2, and into the Type V RuvC domain and activity in general.

-The reported completeness in the PDB validation reports is much lower (around 80-85%) for 6LTP, 6LTR, and 6LTX, while it is close to 100% in Table 1. How can this discrepancy be explained?

The reported completeness in table 1 is the data collection, while the reported completeness in the PDB validation reports is for refinement. As shown in the following table, the overall completeness of our data is 99.3%, and the completeness of each shell is 96.2-100%, displaying the high quality of our data. During the refinement, weak reflections are rejected, resulting in the discrepancy of the completeness from the data collection. In the revision, we further refined our structures and changed the data of structure 6LTR. In our revised version, the overall completeness of all the structures is ~ 90%.

5035 5036 5037 5038 5039 5040 5041 5042 5043 5044 5045 5046 5047 5048 5049 5050 5051 5052 5053 5054 5055 5056 5057 5058 5059 ---	Shell		Summary of observation redundancies:											Completeness total
	Lower limit	Upper limit	% of reflections with given No. of observations											
			0	1	2	3	4	5-6	7-8	9-12	13-19	>19		
	50.00	6.97	1.2	2.5	5.5	7.2	10.8	24.0	36.3	12.7	0.0	0.0	98.8	
	6.97	5.53	0.7	1.1	3.1	5.3	8.0	25.5	40.3	15.9	0.0	0.0	99.3	
	5.53	4.84	1.9	2.1	4.0	5.6	8.7	22.5	38.1	17.2	0.0	0.0	98.1	
	4.84	4.39	3.8	4.2	6.4	8.2	12.7	27.2	29.3	8.3	0.0	0.0	96.2	
	4.39	4.08	2.9	2.5	4.8	6.6	9.3	24.1	37.2	12.7	0.0	0.0	97.1	
	4.08	3.84	2.1	2.3	3.6	5.5	8.8	23.0	40.4	14.3	0.0	0.0	97.9	
	3.84	3.65	1.0	1.6	3.1	5.3	10.1	32.8	34.0	12.1	0.0	0.0	99.0	
	3.65	3.49	0.3	0.8	2.1	3.9	7.0	23.8	44.7	17.4	0.0	0.0	99.7	
	3.49	3.35	0.2	0.6	1.6	3.1	7.0	21.2	47.3	19.0	0.0	0.0	99.8	
	3.35	3.24	0.3	0.2	1.8	1.6	5.4	19.1	50.3	21.4	0.0	0.0	99.7	
	3.24	3.14	0.0	0.4	1.9	2.3	5.9	20.3	46.3	22.9	0.0	0.0	100.0	
	3.14	3.05	0.0	0.6	2.4	2.3	7.2	30.8	39.8	17.0	0.0	0.0	100.0	
	3.05	2.97	0.0	0.2	1.4	2.4	4.7	22.7	49.1	19.4	0.0	0.0	100.0	
	2.97	2.89	0.0	0.2	1.3	1.9	5.9	17.9	51.7	21.1	0.0	0.0	100.0	
	2.89	2.83	0.0	0.1	1.2	1.9	4.9	16.1	51.7	24.1	0.0	0.0	100.0	
	2.83	2.77	0.0	0.0	1.1	1.9	6.0	14.0	52.7	24.3	0.0	0.0	100.0	
	2.77	2.71	0.0	0.0	1.0	1.3	5.6	17.0	49.7	25.3	0.0	0.0	100.0	
	2.71	2.66	0.0	0.3	1.3	3.0	9.1	31.2	39.6	15.6	0.0	0.0	100.0	
	2.66	2.61	0.0	0.2	1.3	2.2	5.7	23.8	48.3	18.5	0.0	0.0	100.0	
	2.61	2.57	0.0	0.2	1.2	2.1	4.9	21.6	48.9	21.2	0.0	0.0	100.0	
	All hkl		0.7	1.0	2.5	3.7	7.4	22.9	43.7	18.0	0.0	0.0	99.3	

-There is a substantial amount of Ramachandran outliers (and a relative high amount of Ramachandran allowed) in each of the structures. Similarly, there is a high amount of rotamer outliers for 6LTP, 6LTR, 6LTU, 6LTX, and 6LU0. Can these Ramachandran and rotamer outliers be explained, or should the structures be refined further?

We thank the reviewer for pointing out this issue. We further refined structures reported in our manuscript, and almost no more residues are in Ramachandran and rotamer outliers.

-Throughout the manuscript, the authors use the terms spacer and repeat for the crRNA. The spacer and repeat are located in the CRISPR locus. Instead, the crRNA contains a spacer-derived segment and a repeat-derived segment. To prevent confusion, the authors should either change this throughout the manuscript (preferred), or indicate once in the beginning of the manuscript (and preferentially in Figure 1) what they mean with the spacer and repeat in the crRNA.

We thank the reviewer for his suggestion. At the beginning of the result section in the revised manuscript, we introduced the repeat-derived segment and spacer-derived segment, which are then referred to as repeat and spacer, respectively.

-It is not clear to me what the source (and/or sequence) of the DNA in the catalytic site is. Can the authors explain this, and explain based on what they have based the sequence in the PDB files? Is this sequence an assumption or can it be deduced with certainty from the density?

The DNA bound in the RuvC catalytic pocket is probably introduced with the target DNA that is synthesized in a company. The nucleotide sequence of the DNA in the catalytic pocket is assumed based on its electron density.

-How do the authors explain that in the structure intact DNA is observed in the catalytic site, while the Cas12i2 is in the cleavage-competent state?

The extremely high concentration of Cas12i2 and the chemicals used for crystallization probably dramatically reduces or inhibits its DNA cleavage activity, resulting an intact DNA substrate bound in the catalytic pocket. A similar phenomena was observed in the *Thermus thermophilus* Argonaute that in complexed with DNA guide and RNA target (PDB ID: 3HVR).

-Can the authors explain if the Helical-II domain and/or the loop that is inserted into the RuvC catalytic pocket is conserved throughout Type V effectors? Has this been described in other papers? For example, can this loop be compared to the 'lid' described for Cas12a (Stella et al. 2018)? If it is conserved, this would not only reveal the activation of Cas12i2, but potentially of other Type V effectors too.

Although the sequences of Helical-II domain and the loop within Helical-II domain of Cas12i2 are not conserved among the type V-Cas12 effectors, Cas12a was reported to be activated by a similar manner, in which the conformational change of the REC2 domain of Cas12a expose the RuvC catalytic site upon the crRNA-TS duplex formation (Swarts, D. 2019). In contrast, the lid described in Stella et al. 2018 is a loop fragment within the RuvC domain. In addition, the helical-II domain of Cas12b was also observed to move away from the Nuc lobe upon the formation guide:target heteroduplex (Yang, 2016). Together, these observation suggest that the activation model of Cas12i2 potentially useful for Cas12a, Cas12b and other Cas12 effectors.

Minor (textual) comments

Below, I have listed various comments that I think will improve readability and will make it easier to understand the figures.

-In my opinion, both the abstract and introduction could do more justice to both earlier findings about Cas12i2 activity (Yan et al.) and to the importance of the work described in this manuscript. The abstract jumps to the knowledge gap and obtained results directly. An introduction would help to put these important findings into context. Furthermore, it is unclear how the seed region and central spacer are defined in the abstract.

We thank the reviewer for his suggestion.

More detail of the Cas12i2 activity and the importance of Yan's work were included in the revised introduction, but not in the abstract due to its 150 words and no reference limitation. In addition, the seed and the central spacer regions are defined in the revised abstract as suggested.

In the introduction, it should be clarified here that Cas12i is most closely related to Cas12b based on phylogeny, but resembles Cas12a in the fact that it also relies on a single guide RNA (and not a tracrRNA). Furthermore, more is known about its cleavage activity (i.e. it has been shown that it does not efficiently cleave ssDNA targets in cis and demonstrates collateral ssDNA cleavage in trans (Yan et al.)) and that should be noted.

Thank the review for this suggestion. The information mentioned by the reviewer was included as suggested.

Page 3:

-Class 2 CRISPR-Cas systems also consist of Cas1, Cas2 and crRNA. It should be rephrased stating that Class 2 rely on a single large protein-crRNA complex for CRISPR interference.

-The authors should explain why RuvC cleaves ssDNA and not dsDNA, as Cas9 and Cas12a both can cleave dsDNA targets.

The sentences have been changed as suggested – "Class 2 CRISPR-Cas systems consist of a single large protein-crRNA complex for CRISPR interference.." and "Cas9, the signature protein of type II system, cleaves the target and non-target strands by its HNH and RuvC catalytic domain. In contrast, Cas12a, the hallmark protein of type V system, use a single RuvC domain for cleavage of both strands of target DNA."

Page 4:

-it should be made clear that Cas12a is expected to cleave the DNA similar to Cas12b (unless the authors disagree based on their findings), as both rely on a single RuvC catalytic site for target cleavage.

We added discussion of the similarity of Cas12a, Cas12b and Cas12 in the revision.

Page 5:

-how identical are the two structures? RMSD is 0.0?

-What is the sequence identity between Cas12i2 and AacCas12b? Are there domains in these two enzymes which are more similar and domains which are less similar?

-Please indicate exactly how many nucleotides of the crRNA and DNA are visible (e.g. 45 out of 54). In fact, it would be insightful to include a supplemental figure displaying all solved structures and the sequences of the RNA and DNA molecules that they have bound, and

color-code whether certain nucleotides in these sequences are visible in the structure or not.

We thank the reviewer for these suggestions. Changes have been made as suggested.

The r.m.s.d between the structure of DNA-bound wild-type Cas12i2 and E833A mutant is 0.3809 Å (1034 residues aligned). The sequence similarity between Cas12i2 and AacCas12b is 21.2%. More comparison between Cas12i2 and AacCas12b was included in the revised version.

Page 8:

-RuvC is not the sole catalytic domain in Cas12i2 – it might be the sole DNA-cleaving catalytic domain, but the authors clearly demonstrate there is a RNA-cleaving catalytic domain too.

The sentence has been changed as suggested -"The RuvC domain is the sole DNase catalytic domain in Cas12i2, with catalytic residues Asp599, Glu833, and Asp1019 lining the active pocket."

Figure 5a:

-The PAM colouring scheme is counterintuitive to me. Can the authors change the colouring scheme or indicate what colours indicates absence and presence of the PAM/mismatches

We clarified the PAM coloring scheme in the figure 5a (Fig. 6a in the revision) caption. The PAM sequence is highlighted in orange, and the DNA without PAM is not highlighted.

REVIEWERS' COMMENTS

Reviewer #1 (Remarks to the Author):

In the revised manuscript, the authors addressed all essential comments. In particular, they performed additional experiments that revealed the residues important for PAM recognition and also revealed that crRNA processing increases dsDNA cleavage. Thus, I am satisfied with the response to my comments.

Reviewer #2 (Remarks to the Author):

Dear editor, dear authors,

In my opinion the authors have done an excellent job in addressing all concerns that I have raised.

I have only two minor recommendations that might aid readers in understanding the data:

- In their rebuttal, the authors give the potential source of the DNA in the catalytic site, and the sequence thereof. This should be mentioned in the manuscript and in Figure 4.
- Similarly, the authors should explain in the manuscript why they believe the substrate is not cleaved while the complex is in the cleavage competent state, as they have explained in their rebuttal.

With kind regards,
Daan C. Swarts

Response to reviewers:

We thank the reviewers for their time and helpful comments on the manuscript. We have made revisions to the manuscript to address their concerns. The responses to the reviewers' comments are below (in blue).

REVIEWER COMMENTS

Reviewer #1 (Remarks to the Author):

In the revised manuscript, the authors addressed all essential comments. In particular, they performed additional experiments that revealed the residues important for PAM recognition and also revealed that crRNA processing increases dsDNA cleavage. Thus, I am satisfied with the response to my comments.

We thank the reviewer for his/her positive comments.

Reviewer #2 (Remarks to the Author):

Dear editor, dear authors,

In my opinion the authors have done an excellent job in addressing all concerns that I have raised.

I have only two minor recommendations that might aid readers in understanding the data: -In their rebuttal, the authors give the potential source of the DNA in the catalytic site, and the sequence thereof. This should be mentioned in the manuscript and in Figure 4.

We thank the reviewer for his suggestion.

The sentences have been changed as suggested – "In the DNA-bound Cas12i2 ternary complexes, a 5-nt ssDNA, which was probably introduced with the target DNA, is observed in the RuvC catalytic pocket".

We also added the sentence "The nucleotide sequence of the ssDNA in the catalytic pocket is assumed based on its electron density." to the caption of Figure 4.

-Similarly, the authors should explain in the manuscript why they believe the substrate is not cleaved while the complex is in the cleavage competent state, as they have explained in their rebuttal.

We thank the reviewer for his suggestion.

The sentence "In addition, the ssDNA bound in the catalytic pocket remains intact may due to the high concentration of the complex and the chemicals used for crystallization," has been added in the revised version.